# Rare disease research workflow using multilayer networks elucidates the molecular determinants of severity in Congenital Myasthenic Syndromes

Iker Núñez-Carpintero [1], Maria Rigau [1,2,3], Mattia Bosio[1,4], Emily O'Connor[5,6], Sally Spendiff[5], Yoshiteru Azuma[7,8], Ana Topf [9,10], Rachel Thompson[5], Peter A. C. 't Hoen [11], Teodora Chamova[12], Ivailo Tournev[12,13], Velina Guergueltcheva[14], Steven Laurie[15], Sergi Beltran [15,16,17], Salvador Capella-Gutiérrez [1,4], Davide Cirillo [1] ✉, Hanns Lochmüller [5,6,15,18,19] & Alfonso Valencia [1,4,20]

Exploring the molecular basis of disease severity in rare disease scenarios is a challenging task provided the limitations on data availability. Causative genes have been described for Congenital Myasthenic Syndromes (CMS), a group of diverse minority neuromuscular junction (NMJ) disorders; yet a molecular explanation for the phenotypic severity differences remains unclear. Here, we present a workflow to explore the functional relationships between CMS causal genes and altered genes from each patient, based on multilayer network community detection analysis of complementary biomedical information provided by relevant data sources, namely protein-protein interactions, pathways and metabolomics. Our results show that CMS severity can be ascribed to the personalized impairment of extracellular matrix components and postsynaptic modulators of acetylcholine receptor (AChR) clustering. This work showcases how coupling multilayer network analysis with personalized -omics information provides molecular explanations to the varying severity of rare diseases; paving the way for sorting out similar cases in other rare diseases.

Understanding phenotypic severity is crucial for prediction of disease outcomes, as well as for administration of personalized treatments. Different severity levels among patients presenting the same medical condition could be explained by characteristic relationships between diverse molecular entities (i.e. gene products, metabolites, etc) in each individual. In this setting, multi-omics data integration is becoming a promising tool for research, as it has the potential to gain complex insights of the molecular determinants underlying disease heterogeneity. However, even in a scenario where the level of biomedical detail available to study is steadily growing[1], the analysis of the molecular determinants of disease severity is not typically addressed in rare disease research literature[2], despite its obvious relevance at the medical and clinical level. Rare diseases represent a challenging setting for the application of precision medicine because, by definition, they affect a small number of patients, and therefore the data available for study is considerably limited in comparison to other conditions. Accordingly, leveraging the wealth of biomedical knowledge of diverse nature coming from publicly available databases has the potential to address data limitations in rare diseases[3,4]. In this sense, multilayer networks can offer a holistic representation of biomedical data resources[5,6], which may allow exploration of the biology related to a

given disease independently of cohort sizes and their available omics data.

Here, in order to evaluate and demonstrate the potential of multilayer networks as means of assessing severity in rare disease scenarios, we provide an illustrative case where we develop a framework for analyzing a patient cohort affected by Congenital Myasthenic Syndromes (CMS), a group of inherited rare disorders of the neuromuscular junction (NMJ). Fatigable weakness is a common hallmark of these syndromes, that affects approximately 1 patient in 150,000 people worldwide. The inheritance of CMS is autosomal recessive in the majority of patients. CMS can be considered a relevant use case because, while patients share similar clinical and genetic features[7], phenotypic severity of CMS varies greatly, with patients experiencing a range of muscle weakness and movement impairment. While over 30 genes are known to be monogenic causes of different forms of CMS (Table 1), these genes do not fully explain the ample range of observed severities, which has been suggested to be determined by additional factors involved in neuromuscular function. Examples of CMS-related genes are *AGRN, LRP4* and *MUSK* encoding for proteins that mediate communication between the nerve ending and the muscle, which is crucial for formation and maintenance of the NMJ (Fig. 1).

In particular, the AGRN-LRP4 receptor complex activates MUSK by phosphorylation, inducing clustering of the acetylcholine receptor (AChR) in the postsynaptic membrane. This allows the presynaptic release of acetylcholine (ACh) to trigger muscle contraction[8,9]. Additional evidence of CMS severity heterogeneity emerged within the NeurOmics and RD-Connect projects[10] studying a small population (about 100 individuals) that were described in the original publication as being of 'gypsy' ethnic origin, from Bulgaria.

All affected individuals shared the same causal homozygous mutation (a deletion within the AChR ε subunit, *CHRNE* c.1327delG[11]). However, the severity of symptoms across this cohort varies considerably regardless of age, gender and initiated therapy, suggesting the existence of additional genetic causes for the diversity of disease phenotypes. By analyzing multi-omics data, we performed an in-depth characterization of 20 CMS patients, representing the two opposite ends of the spectrum observed in the wider cohort, aiming to investigate the molecular bases of the observed differences in the individual severity of the disease. Clinically, CMS severity ranges from minor symptoms (e.g., exercise intolerance) to more severe CMS forms depending on the causal genetic impairments[12,13]. Severe CMS is typically presented with reduced Forced Vital Capacity (FVC), severe generalized muscle fatigue and weakness, proximal and bulbar muscle fatigue and weakness, impaired myopathic gait and hyperlordosis. Two CMS severity levels have been identified for this cohort through extensive phenotyping, namely a severe disease phenotype (8 patients) and a not-severe disease phenotype (2 intermediate and 10 mild patients) (Supplementary Dataset 1). Out of the tested demographic factors (age, sex) and clinical tests (speech, mobility, respiratory dysfunctions, among others), FVC and shoulder lifting ability show a significant association with the severity classes (two-tailed Fisher's exact test $p = 0.0128$ and $p = 0.0418$, respectively; Supplementary Fig. 1). We sought to interrogate whether severity was determined by additional genetic variations impacting neuromuscular activity, on top of the causative *CHRNE* mutation. We analyzed three main types of genetic variations: single nucleotide polymorphisms (SNPs), copy number variations (CNVs), and compound heterozygous variants (two recessive alleles located at different loci within the same gene in a given individual). The extensive analysis of the genomic information did not render any SNPs that could be considered a unique cause of disease severity by being common to all the cases. Nevertheless, a number of CNVs and compound heterozygous variants were found to appear exclusively in the different severity groups, in one or more patients. Moreover, the compound heterozygous variants

of the severe group are enriched in pathways related to the extracellular matrix (ECM) receptors, which have been proposed as a target for CMS therapy[14].

To investigate the functional relationship between these variants and CMS severity, we designed an analytical workflow based on multilayer networks (Fig. 2), allowing the integration of external biological knowledge to acquire deeper functional insights. A multilayer network consists of several layers of nodes and edges describing different aspects of a system[15]. In biomedicine, this data representation has been used to study biomolecular interactions[16] and diseases[6], facilitating

**Table 1 | Location, phenotype, inheritance and genes involved in CMS (adapted from https://omim.org/phenotypicSeries/PS601462 and http://www.musclegenetable.fr)**

| Location | Phenotype | Inheritance | Gene |
|---|---|---|---|
| 2q31.1 | CMS1A, slow-channel | AD | *CHRNA1* |
| 2q31.1 | CMS1B, fast-channel | AR, AD | |
| 17p13.1 | CMS2A, slow-channel | AD | *CHRNB1* |
| 17p13.1 | CMS2C, associated with acetylcholine receptor deficiency | AR | |
| 2q37.1 | CMS3 A, slow-channel | AD | *CHRND* |
| 2q37.1 | CMS3 B, fast-channel | AR | |
| 2q37.1 | CMS3 C, associated with acetylcholine receptor deficiency | AR | |
| 17p13.2 | CMS4 A, slow-channel | AR, AD | *CHRNE* |
| 17p13.2 | CMS4 B, fast-channel | AR | |
| 17p13.2 | CMS4 C, associated with acetylcholine receptor deficiency | AR | |
| 3p25.1 | CMS5 | AR | *COLQ* |
| 10q11.23 | CMS6, presynaptic | AR | *CHAT* |
| 1q32.1 | CMS7, presynaptic | AD | *SYT2* |
| 1p36.33 | CMS8, with pre- and postsynaptic defects | AR | *AGRN* |
| 9q31.3 | CMS9, associated with acetylcholine receptor deficiency | AR | *MUSK* |
| 4p16.3 | CMS10 | AR | *DOK7* |
| 11p11.2 | CMS11, associated with acetylcholine receptor deficiency | AR | *RAPSN* |
| 2p13.3 | CMS12, with tubular aggregates | AR | *GFPT1* |
| 11q23.3 | CMS13, with tubular aggregates | AR | *DPAGT1* |
| 9q22.33 | CMS14, with tubular aggregates | AR | *ALG2* |
| 1p21.3 | CMS15, without tubular aggregates | AR | *ALG14* |
| 17q23.3 | CMS16 | AR | *SCN4A* |
| 11p11.2 | CMS17 | AR | *LRP4* |
| 20p12.2 | CMS18 | AD | *SNAP25* |
| 10q22.1 | CMS19 | AR | *COL13A1* |
| 2q12.3 | CMS20, presynaptic | AR | *SLC5A7* |
| 10q11.23 | CMS21, presynaptic | AR | *SLC18A3* |
| 2p21 | CMS22 | AR | *PREPL* |
| 22q11.21 | CMS23, presynaptic | AR | *SLC25A1* |
| 15q23 | CMS24, presynaptic | AR | *MYO9A* |
| 12p13.31 | CMS25, presynaptic | AR | *VAMP1* |
| 3p21.31 | CMS, related to GMPPB | AR | *GMPBB* |
| 20q13.33 | CMS, presynaptic | AR | *LAMA5* |
| 3p21.31 | CMS, with nephrotic syndrome | AR | *LAMB2* |
| 8q24.3 | CMS, with plectin defect | AR | *PLEC* |
| 12q24.13 | CMS, related to RPH3A | AR | *RPH3A* |
| 9p13.3 | CMS, presynaptic, related to MUNC13-1 | AR | *UNC13B* |
| 2q37.1 | Escobar syndrome | AR | *CHRNG* |

*AR* autosomal recessive, *AD* autosomal dominant.

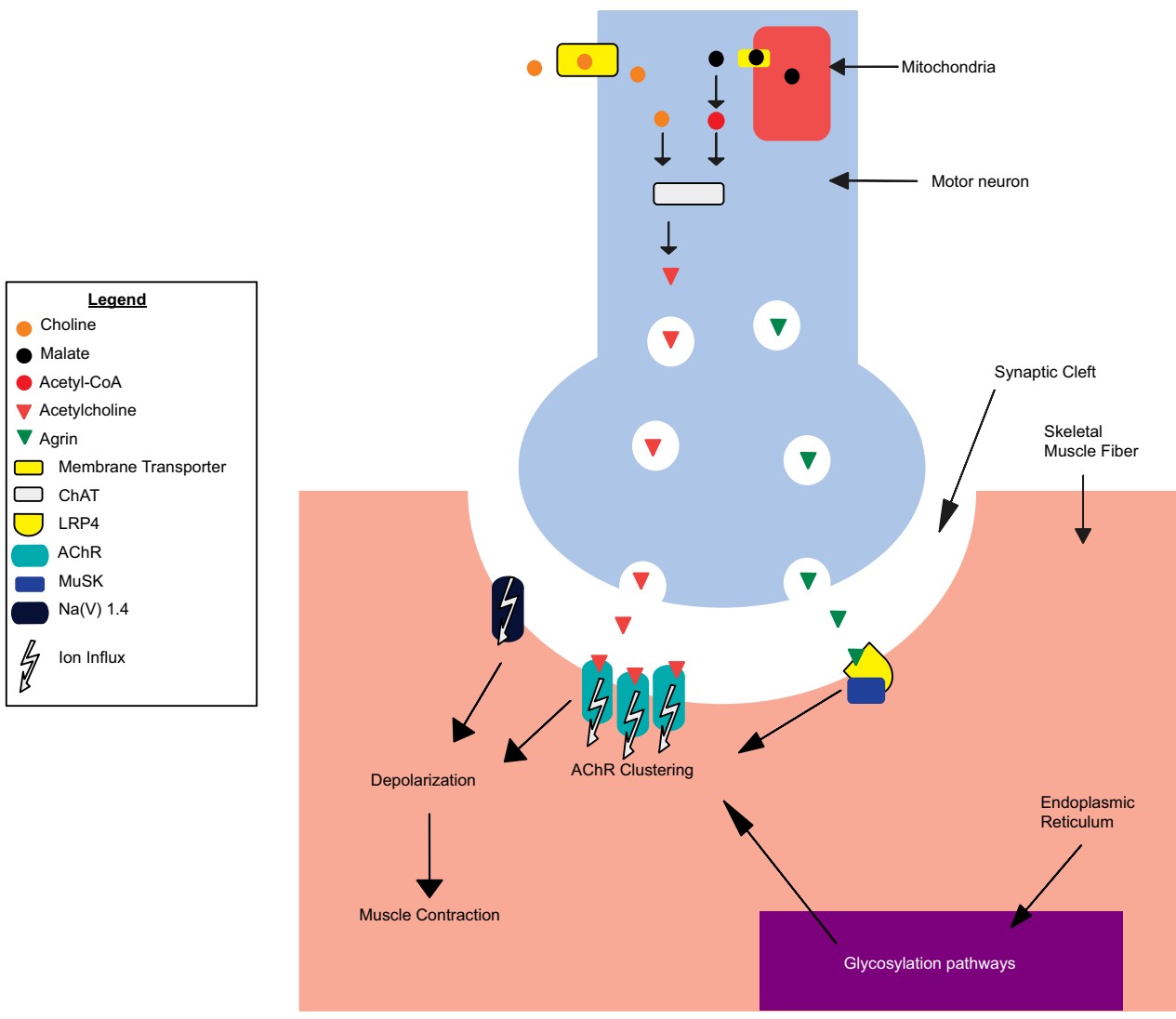

**Fig. 1 | A schematic depiction of the main molecular activities of known Congenital Myasthenic Syndromes (CMS) causal genes (Methods) taking place at the neuromuscular junction (NMJ) in the presynaptic terminal (in blue), synaptic cleft (in white), and skeletal muscle fiber (in red) (for a detailed description of this system see Supplementary Information, Functions of CMS-** associated genes in the neuromuscular junction). ChAT Choline O-Acetyl-transferase, LRP4 LDL Receptor Related Protein 4, AChR Acetylcholine Receptor, MuSK Muscle Associated Receptor Tyrosine Kinase, Na(V) 1.4 Na$_v$1.4 voltage-gated sodium channel.

integration and interpretation of heterogeneous sources of data. Several established tools for network analysis have been recently adapted for multilayer networks, such as random walk with restart[17,18], community detection algorithms[19] and node embeddings[20]. By crossing patient genomic data with the information provided by a multilayer network encompassing biomedical knowledge, we are able to describe the functional relationships of new genetic modifiers responsible for the different phenotypic severity levels, showcasing the potential of multilayer networks to provide support on the analysis of rare disease patients.

## Results

### Variants do not segregate with patient severity

We first searched for variants able to segregate the disease phenotypes (severe and not-severe) by analyzing a large panel of mutational events (mutations in isoforms, splicing sites, small and long noncoding genes, promoters, transcription start site (TSS), predicted pathogenic mutations, loss of function mutations, among others). We could not find one single mutation or combinations of mutations that were able to completely segregate the two groups (Supplementary Information,

Supplementary Fig. S1) although partial segregation can be observed (Suppl. Dataset 2). As already described for monogenic diseases[21] and cancer[22], we hypothesized that distinct weak disease-promoting effects may represent patient-specific causes to CMS severity, which bring damage to sets of genes that are functionally related. To find these effects, we sought to search for variants with the potential to alter gene functions, such as CNVs and compound heterozygous variants, which have been previously reported to be key to CMS[12,23–25].

### Compound heterozygous variants are functionally related

In order to explore the hypothesis that disease severity in this cohort may be due to variants in patient-specific critical elements, we sought to identify potentially damaging compound heterozygous variants and CNVs. We analyzed the gene lists associated with these mutations to search for evidence of alterations in relevant pathways for the severe ($n = 8$) and not-severe cases ($n = 12$). We first performed a functional enrichment analysis (Methods) of the genes with CNVs found in the two groups. The set of affected genes in the severe group is composed of 26 unique genes (10 private to the severe group), while the not-severe group presented 86 unique genes (Supplementary Dataset 3).

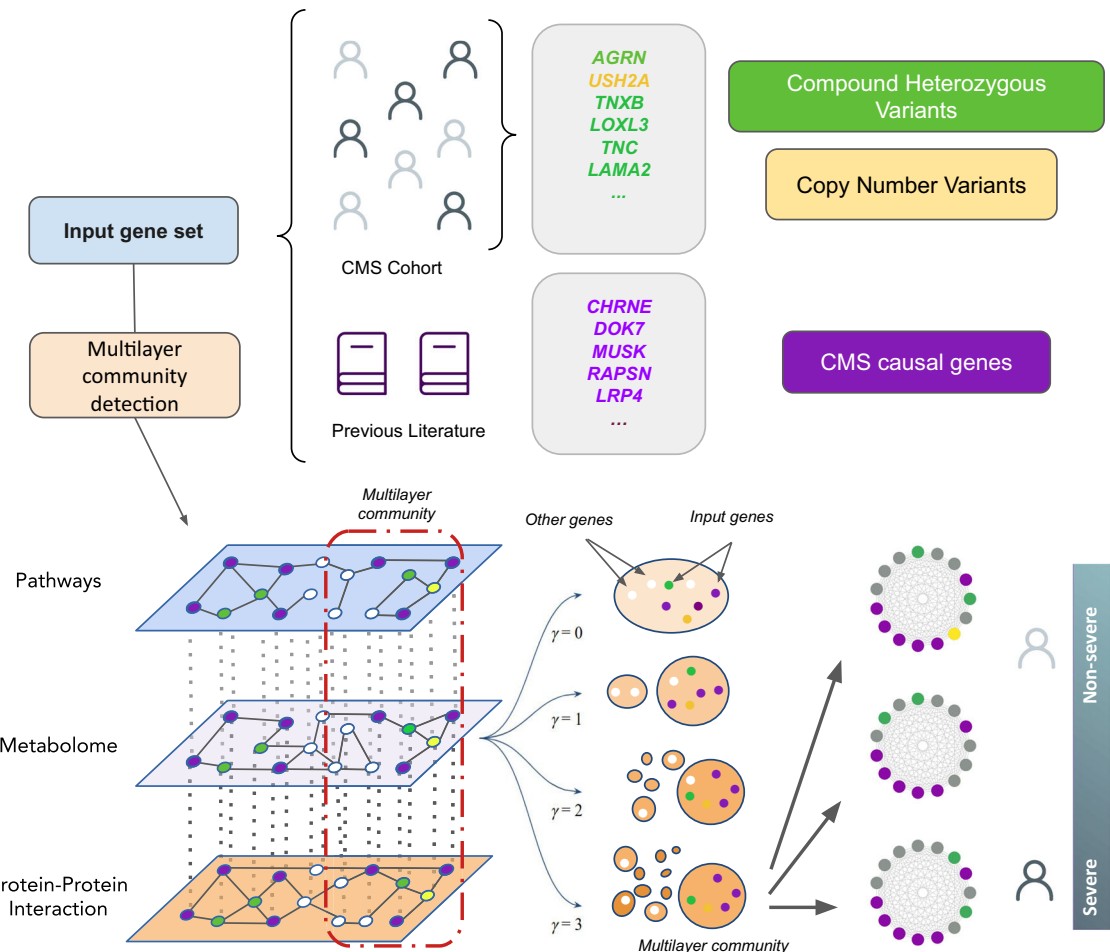

**Fig. 2 | Analytical workflow designed to address the severity of a cohort of patients affected by Congenital Myasthenic Syndromes (CMS).** A multi-scale functional analysis approach, based on multilayer networks, was used to identify the functional relationships between genetic alterations obtained from omics data (Whole Genome Sequencing, WGS; RNA-sequencing, RNAseq) with known CMS causal genes. In green, compound heterozygous variants; in yellow, copy number variants (CNVs); in purple, known CMS causal genes. Modules of CMS linked genes are detected using graph community detection at a resolution range ($\gamma$) (Methods) where the most prominent changes in community structure occur. Modules that emerged from this analysis were characterized at single individual level.

None of these gene sets showed any functional enrichment. Moreover, none of these genes had been described as causal for CMS, and none carried compound heterozygous variants (Supplementary Fig. 2). As for compound heterozygous variants, the set of affected genes in the severe group is composed of 112 unique genes (89 private to the severe group), while the not-severe group resulted in 152 unique genes (Supplementary Dataset 3). We found that the severe group shows significant enrichment in genes belonging to extracellular matrix (ECM) pathways, in particular ECM receptor interactions (KEGG hsa04512, $p$ adjusted = 0.002337) and ECM proteoglycans (Reactome R-HSA-30001787, $p$ adjusted = 0.001237), which are the top-hit pathways when the 89 genes appearing only in the severe group are considered. Both these pathways share common genes, namely *TNXB*, *LAMA2*, *TNC*, and *AGRN*. The role of extracellular matrix proteins for the formation and maintenance of the NMJ has recently drawn attention to the study of CMS[26,27]. In particular, within the genes linked with ECM pathways, *AGRN* and *LAMA2* stand out for their implication in CMS and other rare neuromuscular diseases[28–30]. ECM-related pathways are not enriched in the not-severe set of genes (KEGG hsa04512, $p$ adjusted = 0.6170). Moreover, top-hit pathways of the not-severe set of genes are not explicitly related to ECM and not consistent between Reactome and KEGG (Reactome Susceptibility to colorectal cancer R-HSA-5083636, $p$ adjusted = 4.131e−7, genes *MUC3A/5B/12/16/17/19*; KEGG Huntington's disease hsa05016, $p$ adjusted = 0.07103, genes *REST, CREB3L4, CLTCL1, DNAH2/8/10/11*). These findings support our

hypothesis that the severe patients might present disruptions in NMJ functionally related genes that, combined with *CHRNE* causative alteration, may be responsible for the worsening of symptoms.

## CMS-specific monolayer and multilayer community detection

As disease-related genes tend to be interconnected[31], we sought to analyze the relationships among the CMS linked genes (i.e. known CMS causal genes, and severe and not-severe compound heterozygous variants and CNVs; Methods) using network community clustering analysis. We employed the Louvain algorithm (Methods) to find groups of interrelated genes in three monolayer networks that represent biological knowledge contained in databases, separately: the Reactome database[32], the Recon3D Virtual Metabolic Human database[33], and from the Integrated Interaction Database (IID)[34] (Supplementary Fig. 3). The first network consists of 10,618 nodes (genes) and 875,436 edges, representing shared pathways between genes. The second network consists of 1863 nodes (genes) and 902,188 edges, representing shared reaction metabolites between genes. The third network consists of 18,018 nodes (genes) and 947,606 edges, representing aggregated protein-protein interactions from all tissues (Methods: Monolayer community detection). The last two networks, represent the 'metabolome' and the 'interactome' data, respectively. Measurement of network overlap and community similarity (Methods) revealed high specificity of their edges, as well as that the same CMS linked genes did not form

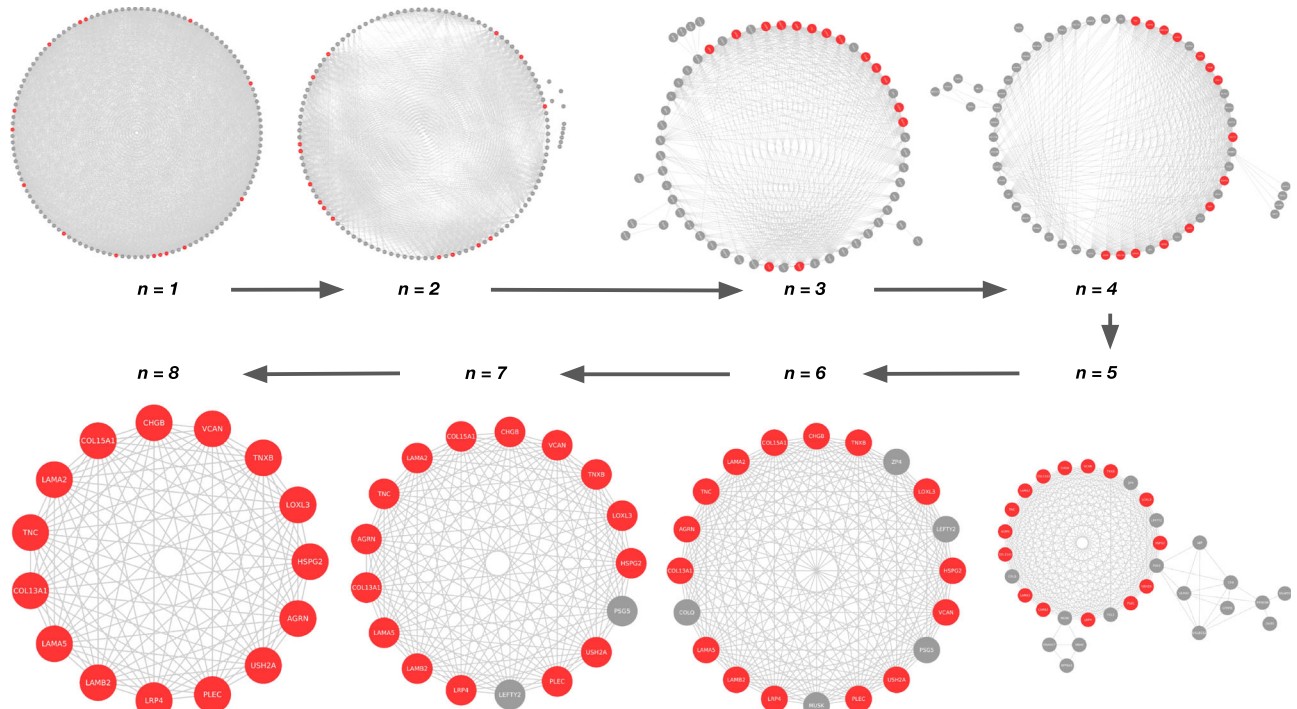

**Fig. 3 | Identification of the largest module containing genes that are found in the same community in a range of modularity resolution (Methods).** In each module, genes are connected if they are found in the same multilayer communities at $n$ values of the resolution parameter $\gamma$ within the range under consideration ($\gamma \in (0,4]$). The arrows indicate the systematic increase of $n$. At $n = 8$, the module contains genes that are always found in the same community in the entire range of resolution (see Supplementary Information, Multilayer community detection analysis). The largest module containing the CMS linked gene set (highlighted in red), which includes known CMS causal genes, severe-specific heterozygous compound variants and CNVs, is shown. Source data are provided in the Github repository of the project (see Data Availability section).

the same communities across the different networks (Supplementary Fig. 4).

These results show that, although disease-related genes are prone to form well-defined communities in distinct networks[35,36], different facets of biological information reflect diverse participation modalities of such genes into communities. In order to deliver an integrated analysis of such heterogeneous information, we further consider them as a multilayer network[5] (Methods: Monolayer community detection and Multilayer community detection).

### Large-scale multilayer community detection of disease associated genes

We first sought to test the hypothesis that disease-related genes tend to be part of the same communities also in a multilayer network setting. We used the curated gene-disease associations database DisGeNET[37], showing that disease-associated genes are significantly found to be members of the same multilayer communities (Wilcoxon test $p < 0.001$ in a range of resolution parameters described in the Methods). We pre-processed DisGeNET database by filtering out diseases and disease groups with only one associated gene (6352 diseases), and those whose number of associated genes was more than 1.5 * interquartile range (IQR) of the gene associated per disease distribution (823 diseases with more than 33 associated genes) (Supplementary Fig. 5A, B). This procedure prevents a possible analytical bias due to the higher amounts of genes annotated to specific disease groups (e.g. entry C4020899, Autosomal recessive predisposition, annotates 1445 genes). We then retrieved the communities of each associated gene, excluding 428 genes not present in our multilayer network and the diseases left with only one associated gene. The final analysis comprised a total of 5892 diseases with an average number of 7.38 genes per disease.

For each disease, we counted the number of times that disease-associated genes are found in the same multilayer communities, and compared the distribution of such frequencies with that of balanced random associations (1000 randomizations). Results show that disease-associated genes are significantly found in the same multilayer communities across the resolution interval (Suppl. Figure 5C).

### Modules within the CMS multilayer communities

We define a module as a group of CMS linked genes that are systematically found to be part of the same multilayer community while increasing the multilayer network community resolution parameter (Methods; Supplementary Information, Supplementary Fig. S2; Figs. 3 and 4).

Within each of these communities, we identified smaller modules of CMS linked genes that are specific to the severe and not-severe groups. We tested the significance of obtaining these exact genes in the severe and not-severe largest modules upon severity class label shuffling among all individuals (1000 randomizations). We found that 13 ($p$ adjusted = 0.022) and 14 ($p$ adjusted = 0.027) are the minimum number of genes composing the modules that are not expected to be found at random in the severe and not-severe largest components, respectively (Supplementary Fig. 6).

In the two groups, the significantly largest module that contains known CMS causal genes is composed of 15 genes (Fig. 4). 6 out of these 15 are previously described CMS causal genes (Methods), namely the ECM heparan sulfate proteoglycan agrin (*AGRN*); the cytoskeleton component plectin (*PLEC*), causative of myasthenic disease[38]; the agrin receptor *LRP4*, key for AChR clustering at NMJ[39] and causative of CMS by compound heterozygous variants[40]; the ECM components *LAMA5* and *LAMB2* laminins, and *COL13A1* collagen. Considering all nodes (not only CMS linked), the number of nodes in the module is 482. All the

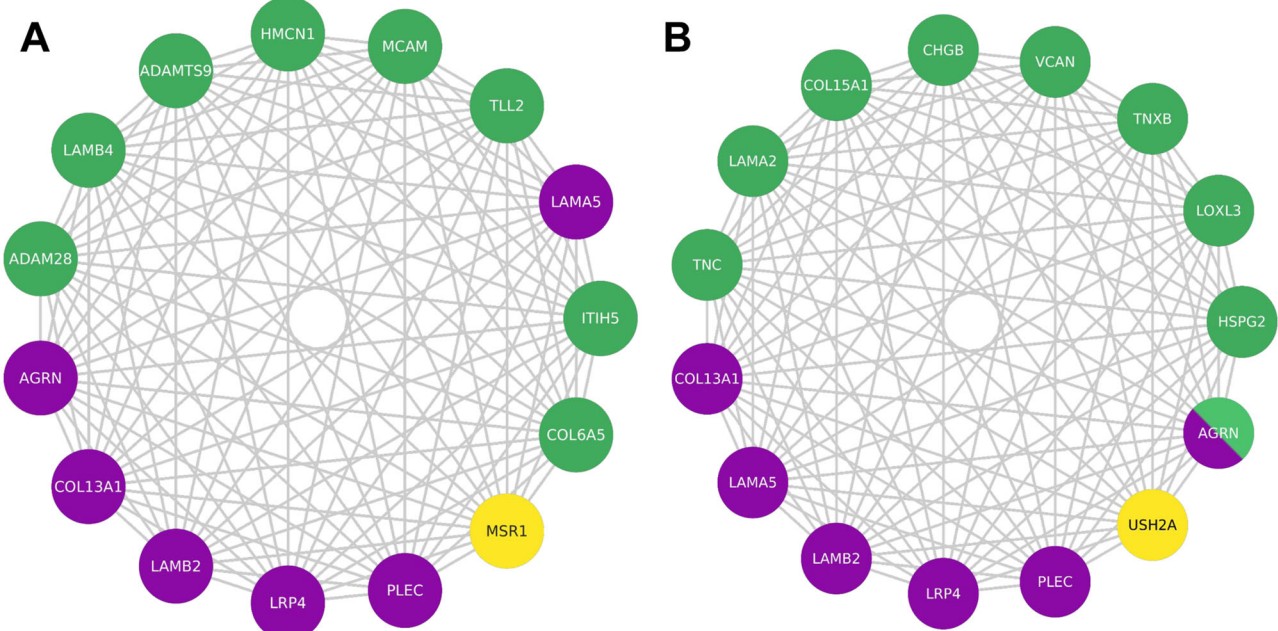

**Fig. 4 | Largest multilayer network modules containing known CMS causal genes.** The largest modules, containing known CMS causal genes, within the multilayer communities of CMS linked genes specific to the not-severe (**A**) and severe (**B**) groups are reported. In green, compound heterozygous variants; in yellow, CNVs; in purple, known CMS causal genes. Being a CMS causal gene bearing compound heterozygous variants, *AGRN* is depicted using both green and purple. Source Cytoscape session is provided in the Github repository of the project (see Data Availability section).

other genes of the two modules are involved in a varied spectrum of muscular dysfunctions, discussed in the following sections. As the location of the causal gene products determine the most common classification of the disease (i.e. presynaptic, synaptic, and post-synaptic CMS)[27], we determined class and localization of the members of the found modules (Table 2).

Laminins, well-known CMS glycoproteins, are affected in both severe (*LAMA2*, *USH2A*) and not-severe (*LAMB4*) groups, and are bound by specific receptors that are damaged in the not-severe group (*MCAM*)[41]. Collagens, known CMS-related factors, are associated with the not-severe group (*COL6A5*), and bound by specific receptors that are damaged in the not-severe group (*MSR1*)[42].

However, overall collagen biosynthesis is affected in both severe and not-severe groups. Indeed, metalloproteinases, damaged in the not-severe group, are responsible for the proteolytic processing of lysyl oxidases (*LOXL3*), which are implicated in collagen biosynthesis[43] and damaged in the severe group. Alterations in proteoglycans (*AGRN*, *HSPG2*, *VCAN*, *COL15A1*)[44], tenascins (*TNC*, *TNXB*)[45,46], and chromogranins (*CHGB*)[47] are specific of the severe group. We observed no genes associated with proteoglycan damage in the not-severe group, suggesting a direct involvement of ECM in CMS severity.

### Personalized analysis of the severe cases
We sought to analyze the 15 genes of the largest module of the severe group in each one of the 8 patients, hereafter referred to using the WGS sample labels (Supplementary Dataset 1). At the topological level, all incident interactions existing between the genes of the severe module (Fig. 4B) are related to the protein-protein interaction and pathway layers (Fig. 5). Overall, these genes have a varied range of expression levels in tissues of interest (Supplementary Fig. 7), for instance in skeletal muscle *HSPG2*, *LAMA2*, *PLEC* and *LAMB2* show medium expression levels (9 to 107 TPM) while the others show low expression levels (0.6 to 9 TPM) (Methods). Patient 2, a 15 years old male, presents compound heterozygous variants in tenascin C (*TNC*), mediating acute ECM response in muscle damage[45,48], and CNVs (specifically, a partial heterozygous copy number loss) in usherin

(*USH2A*), which have been associated with hearing and vision loss[49]. Patient 16, a 25 years old female, presents compound variants in tenascin XB (*TNXB*), which is mutated in Ehlers-Danlos syndrome, a disease that has already been reported to have phenotypic overlap with muscle weakness[50–53] and whose compound heterozygous variants have been reported for a primary myopathy case[54,55]; and versican (*VCAN*), which has been suggested to modify tenascin C expression[56] and is upregulated in Duchenne muscular dystrophy mouse models[57,58]. Patient 13, a 26 years old male, presents compound mutations in laminin α2 chain (*LAMA2*), a previously reported gene related to various muscle disorders[59–61] whose mutations cause reduction of neuromuscular junction folds[62], and collagen type XV α chain (*COL15A1*), which is involved in guiding motor axon development[63] and functionally linked to a skeletal muscle myopathy[64,65]. Patient 12, a 49 years old female, presents compound mutations in chromogranin B4 (*CHGB*), potentially associated with amyotrophic lateral sclerosis early onset[66,67]. Patient 18, a 51 years old man, presents compound mutations in agrin (*AGRN*), a CMS causal gene that mediates AChR clustering in the skeletal fiber membrane[68,69]. Patient 20, a 57 years old male, presents compound mutations in lysyl oxidase-like 3 (*LOXL3*), involved in myofiber extracellular matrix development by improving integrin signaling through fibronectin oxidation and interaction with laminins[70], and perlecan (*HSPG2*), a protein present on skeletal muscle basal lamina[72,73], whose deficiency leads to muscular hypertrophy[74], that is also mutated in Schwartz-Jampel syndrome[75], Dyssegmental dysplasia Silverman-Handmaker type (*DDSH*)[76] and fibrosis[77], such as Patient 19, a 62 years old female. Furthermore, based on the estimated familial relatedness (Methods) and personal communication (February 2018, Teodora Chamova), patients 19 and 20 are siblings (Supplementary Dataset 4).

### Functional consequences of variants in the severe-specific module
Studying the functional impact of the compound heterozygous variants in the severe-specific genes of the module, we observed that in 6 of the 8 patients at least one of the variants is predicted to be

**Table 2 | Localization and functions of proteins encoded by the genes found in the largest modules of the multilayer communities of severe and not-severe groups**

| Activity localization | Class | CMS causal gene | Phenotype group | | Function | Synaptic localization (Manual curation) | Localization (UniProt) |
|---|---|---|---|---|---|---|---|
| | | | Not-severe | Severe | | | |
| ECM (ECM) | Proteoglycans | AGRN | – | AGRN | Cell hydration and growth factor trapping | Pre- and postsynaptic (PMID: 29462312) | Synaptic basal lamina/ECM |
| | | – | – | HSPG2 | | Basement membrane (PMID:30453502) | Basement membrane/ECM |
| | | – | – | VCAN | | ECM (PMID:29211034) | ECM |
| | | – | – | COL15A1 | | Basement membrane (PMID:26937007) | ECM |
| | Collagens | COL13A1 | – | – | Structural support | Basement membrane (PMID: 30768864) | Post-synaptic cell membrane |
| | | – | COL6A5 | – | | Basement membrane (PMID:23869615) | Extracellular matrix |
| | Laminins | LAMA5 | – | – | Web-like structures | Pre-synaptic (PMID:28544784) | Basement membrane/ECM |
| | | LAMB2 | – | – | | Basement membrane (PMID:27614294) | Basement membrane/ECM/Synaptic cleft |
| | | – | LAMB4 | – | | Myenteric plexus basement membrane (PMID: 28595269) | Basement membrane/ECM |
| | | – | – | LAMA2 | | Pre-synaptic (PMID:9396756) | Basement membrane/ECM |
| | | – | – | USH2A | | Neuronal projection of stereocilia (PMID:19023448) | Stereocilia membrane/Secreted (Extracellular region) |
| | Fibulins | – | HMCN1 | – | Scaffolding | Glomerular Extracellular matrix (PMID: 29488390) | Basement membrane/ECM |
| | Tenascins | – | – | TNC | Anti-adhesion | Basement membrane (PMID: 29466693) | ECM/Perisynaptic ECM (Ensembl) |
| | | | | TNXB | | Basement membrane (PMID: 23768946) | ECM |
| | | | | LOXL3 | Collagen assembly | Basement membrane (PMID:26954549) | Secreted (extracellular region) |
| | | ADAMTS9 | – | | Proteoglycan cleavage | Secreted to ECM (PMID:30626608) | ECM |
| | | ADAM28 | – | | | ECM (PMID:24613731) | Cell membrane/Secreted (extracellular region) |
| | Neuropeptides | – | – | CHGB | Regulatory peptides precursor | Pre- and postsynaptic (PMID:7526287) | Secreted (extracellular region) |
| | Others | – | ITIH5 | – | Hyaluronic acid binding | ECM (PMID:27143355) | Secreted (extracellular region) |
| Cell surface | Receptors | – | MSR1 | – | Proteoglycan and collagen binding | Macrophage surface Scavenger Receptor (PMID:12488451) | Plasma membrane |
| | | | MCAM | | | Plasma membrane (PMID:28923978) | Plasma membrane |
| | | LRP4 | – | – | Laminin binding | Post-synaptic (PMID:25319686) | Post-synaptic cell membrane |
| Cytoplasm | Cytoskeleton | PLEC | – | – | Structural support | Post-synaptic (PMID:20624679) | Post-synaptic cytoskeleton |

Synaptic localization was retrieved from manual curation and Uniprot database (Methods).

deleterious by the Ensembl Variant Effect Predictor (VEP)[78] (Methods; Supplementary Dataset 5). For example, as for Patient 18, who presents 3 different variants in *AGRN* gene, only rs200607541 is predicted to be deleterious by VEP's Condel (score = 0.756), SIFT (score = 0.02), and PolyPhen (score = 0.925). In particular, the variant (C > T transition) presents an allele frequency (AF) of 4.56E−03 (gnomAD exomes)[79] and affects a region encoding a position related to a EGF-like domain (SMART:SM00181) and a Follistatin-N-terminal like domain

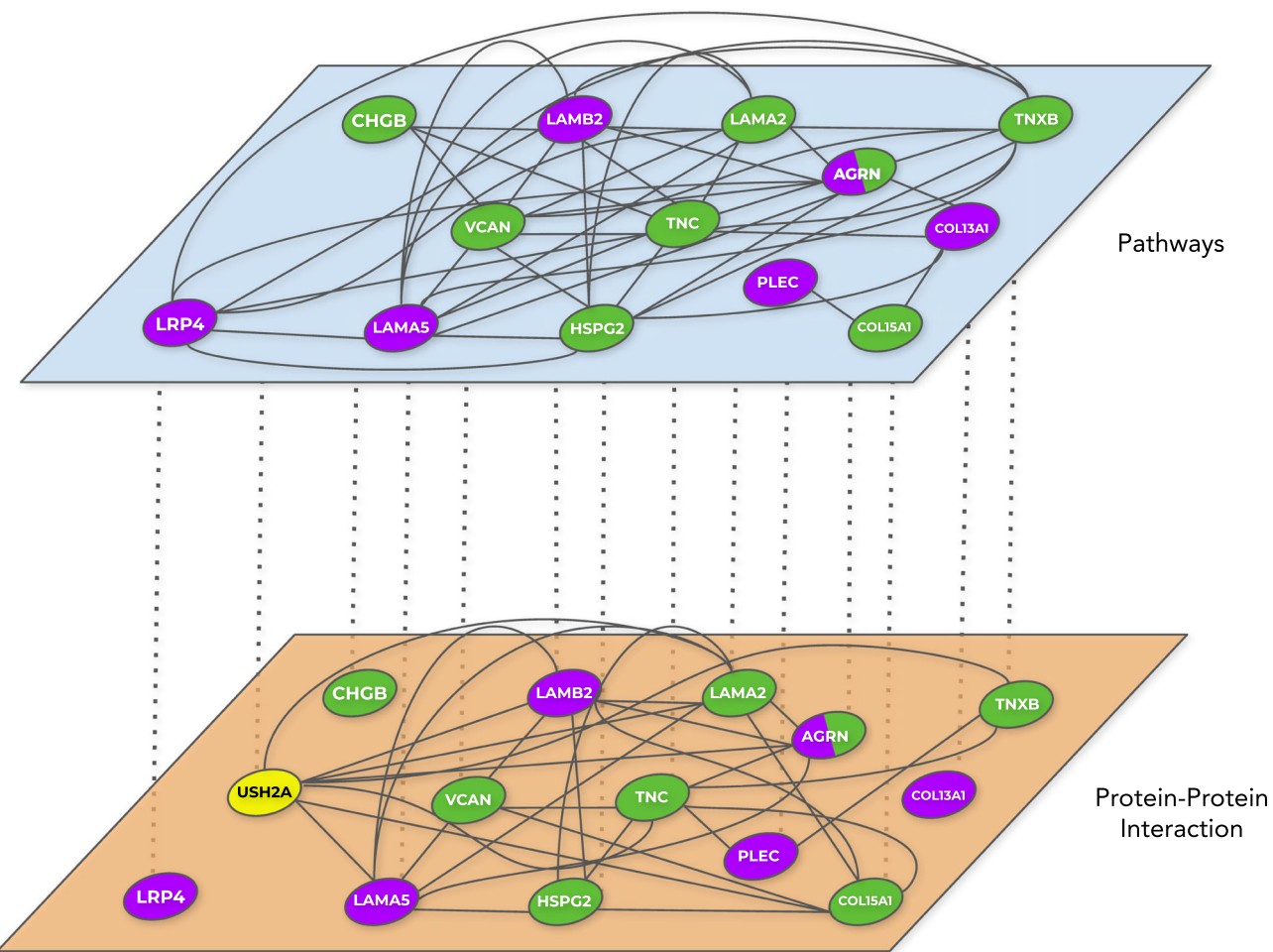

**Fig. 5 | Incident interactions between the genes identified in the severe-specific module in the multilayer network.** LOXL3 is not depicted as it has incident interactions with genes in the module that are not CMS linked. USH2A is not present in the pathways layer, thus it is only depicted in the protein-protein interaction layer.

(SMART:SM00274). Both of these domains are part of the Kazal domain superfamily which is specially found in the extracellular part of agrins (PFAM: CL0005)[80,81]. On the other hand, Patient 16 presents a total of 38 *TNXB* transcripts affected by three gene variants (rs201510617, rs144415985, rs367685759) that are all predicted to be deleterious by the three scoring systems, have allele frequencies of 3.17E−02, 4.83E−02 and 5.90E−03, respectively; and in overall, are affecting two conserved domains. The first consists of a fibrinogen related domain that is present in most types of tenascins (SMART:SM00186), while the second is a fibronectin type 3 domain (SMART:SM00060) that is found in various animal protein families such as muscle proteins and extracellular-matrix molecules[82]. Two of the severe patients (Patients 12 and 19) present severe-only specific compound heterozygous variants that are not predicted to be deleterious. However, one variant in the *CHGB* gene (rs742710, AF = 1.07E −01), present in patient 12, has been previously reported to be potentially causative for amyotrophic lateral sclerosis early onset[66,67]. This gene has also been strongly suggested in literature as a possible marker for onset prediction in multiple sclerosis[83], and other related neural diseases like Parkinson's[84] and Alzheimer's disease[85]. As for Patient 19, the variant rs146309392 (AF = 8.40E−04) in the gene *HSPG2* has been previously referred to be causal of Dyssegmental dysplasia as a compound heterozygous mutation[76]. This variant, as pointed out before, is shared with sibling patient 20. One severe individual (Patient 3), a 37 years old female, does not carry compound heterozygous variants included in this module but others at a lower resolution

parameter value (Supplementary Fig. 8; Supplementary Dataset 6). Interestingly, most of the genes carrying severe-specific deleterious compound heterozygous variants in this patient (*CDH3, FAAP100, FCGBP, GFY, RPTN*) are not related to processes at the NMJ level[86–90]. Nevertheless, three of these variants occur in genes potentially involved in NMJ functionality. In particular, variants rs111709242 (AF = 2.64E−03) and rs77975665 (AF = 3.03E−02) affect gene *PPFIBP2*, which encodes a member of the liprin family (liprin-β) that has been described to control synapse formation and postsynaptic element development[91,92]. Furthermore, the variant rs111709242 is predicted to be deleterious by the SIFT algorithm (see Supplementary Dataset 6). Interestingly, PPFIBP2 appears in modules at lower resolution parameter values associated with known CMS causal genes (e.g. *DOK7, RPSN, RPH3A, VAMP1, UNC13B*) (Supplementary Fig. 8). In addition, variant rs151154986 (AF = 2.18E−02) affects the acyl-CoA thioesterase *ACOT2*, which generate CoA and free fatty acids from acyl-CoA esters in peroxisomes[93]. While *ACOT2* is not retained across the entire resolution range explored, community detection at the individual layer level (i.e. Louvain community detection for each network) revealed relationships with causal CMS genes at all layers (Supplementary Fig. 3). Namely, *ACOT2* shares community membership with *ALG14, DPAGT1, GFPT1, GMPPB* and *SLC25A1*A at the protein-protein interaction layer; with *CHAT* and *SLC5A7* at the pathways layer; and with *GMPBB, SLC25A1* and *CHAT* at the metabolomic layer. A role for CoA levels in skeletal muscle for this enzyme class has been previously described[94]. Moreover, this patient presents high relatedness with

three not-severe patients (Patients 8, 9, and 10) who in turn display a very high relatedness among them (Supplementary Dataset 4).

## Potential pharmacological implications

Finding a personalized genetic diagnosis might help select the appropriate medication for each patient. For instance, fluoxetine and quinine are used for treating the slow-channel syndrome, an autosomal dominant type of CMS caused by mutations affecting the ligand binding or pore domains of AChR, but this treatment should be avoided in patients with fast-channel CMS[95]. Within our cohort, 13 (7 mild, 2 moderate and 4 severe) out of 20 individuals from our CMS cohort are receiving a pharmacological treatment consisting of pyridostigmine, an acetylcholinesterase inhibitor used to treat muscle weakness in myasthenia gravis and CMS[96]. This treatment slows down acetylcholine hydrolysis, elevating acetylcholine levels at the NMJ, which eventually extends the synaptic process duration when the AChR are mutated. Although the severity could potentially be related to how well a patient responds to the treatment with the AchE inhibitors, we could not find a clear correlation between severity and pyridostigmine treatment (two-tailed Fisher's exact test $p$ adjusted = 0.356; Supplementary Fig. 1). In Addition to the causal mutation in *CHRNE*, our results indicate that severity is related to AChR clustering at the Agrin-Plectin-LRP4-Laminins axis level, suggesting the potential benefit of pharmaceutical intervention enhancing the downstream process of AChR clustering. For example, beta-2 adrenergic receptor agonists like ephedrine and salbutamol have been documented as capable of enhancing AChR clustering[97] and proved to be successful in the treatment for severe AChR deficiency syndromes[98,99]. Furthermore, the addition of salbutamol in pyridostigmine treatments has been described as being able to ameliorate the secondary effects of pyridostigmine in the postsynaptic structure[100].

## Discussion

In this work, we have developed a framework for the analysis of disease severity in scenarios heavily impacted by sample size. Presenting limited numbers of cases is one of the main obstacles for the application of precision medicine methods in rare disease research, as it critically affects the level of expected statistical power, a common hallmark in the analysis of minority conditions[101]. This fact hampers exploring the molecular relationships that define the inherently heterogeneous levels of disease severity observed in rare disease populations, making it an atypically addressed biomedical problem[2]. Our approach, based on the application of multilayer networks, enable the user to account for the many interdependencies that are not properly captured by a single source of information, effectively combining the available patient genomic information with general biomedical knowledge from relevant databases representing different aspects of molecular biology. The application to a relevant clinical case, where we tested the hypothesis that the severity of CMS may determined by patient-specific alterations that impact NMJ functionality, provided evidence on how the methodology is able to recover the molecular relationships between the candidate patient-specific genomic variants, the observed causal AChR mutation and previously described CMS causal genes (Table 1).

Our in-depth functional analysis focused on a cohort of 20 CMS patients, from a narrow, geographically isolated and ethnically homogenous population, who share the same causative mutation in the AChR ε subunit (*CHRNE*) but show different levels of severity. The isolation and endogamy that characterize the population from which these patients come from might have favored the accumulation of damaging variants[102,103], giving rise to the emergence of compound effects on relevant genes for CMS. This observation has previously been made in similar syndromes[104,105] and in a number of other neuromuscular diseases[106,107]. Compound heterozygosity is known to happen in CMS[108,109]. The initial analysis of compound heterozygous

variants revealed a significant enrichment of functional categories that are specific to the severe cases, namely ECM functions. This suggests the existence of functional relationships between major actors of the NMJ that are affected by severity-associated damaging mutations. Such interactors include already known CMS causal genes (e.g. *AGRN, LRP4, PLEC*) as well as genes known to interact with them. While severity-specific compound heterozygous variants and CNVs are observed, demographic factors (e.g. sex, age), pharmacological treatment, and personalized omics data (e.g. variant calling, differential gene expression, allele specific expression, splicing isoforms) do not segregate with patient severity.

Therefore, this motivated the development of our multilayer network community analysis to investigate the relationship between known CMS causal genes and severity-associated variants (compound heterozygous variants and CNVs), integrating pathways, metabolic reactions, and protein-protein interactions. Recently, we used a multilayer network as a means to perform dimensionality reduction tasks for patient stratification in medulloblastoma, a childhood brain tumor[110]. Here, we started by analyzing DisGeNET data in order to verify that disease-associated genes tend to belong to the same multilayer communities. We then identified stable and significantly large gene modules within our CMS cohort's multilayer communities and mapped the corresponding damaging mutations back to the single patients, providing a personalized mechanistic explanation of severity differences. Given the difficulties of cohort recruitment for rare diseases, this approach could be used to investigate forms of CMS and other phenotypically variable rare diseases caused by a common mutation.

Overall, our approach revealed major relationships at the protein-protein and pathway layers. The personalized analysis of these mutations further suggests that CMS severity can be ascribed to the damage of specific molecular functions of the NMJ which involve genes belonging to distinct classes and localizations, namely ECM components (proteoglycans, tenascins, chromogranins) and postsynaptic modulators of AChR clustering (*LRP4, PLEC*) (Table 2). Alterations of other genes related to ECM components, such as laminins and collagen, are observed but are not specific to the severity levels.

Although at first the use of metabolomic knowledge in the multilayer network did not seem to provide highly relevant information for the cohort, it provided relevant insights for the personalized analysis of Patient 3, whose mutations presented functional relationships in all layers with other CMS causal genes outside of the presented severe-specific module (Supplementary Fig. 3).

Finding a personalized genetic diagnosis for phenotypic severity might help select the appropriate medication for each patient. Within our cohort, 13 out of 20 individuals from our CMS cohort are receiving a pharmacological treatment consisting of pyridostigmine, an acetylcholinesterase inhibitor used to treat muscle weakness in myasthenia gravis and CMS[96]. Although the severity could potentially be related to how well a patient responds to the standard treatment with the AchE inhibitors, we could not find a clear correlation between severity and pyridostigmine treatment (two-tailed Fisher's exact test $p$ adjusted = 0.356; Supplementary Fig. 1). Our results indicate that severity is related to AChR clustering at the Agrin-Plectin-LRP4-Laminins axis level, suggesting the potential benefit of pharmaceutical intervention enhancing the downstream process of AChR clustering. Strikingly, beta-2 adrenergic receptor agonists like ephedrine and salbutamol have been documented as capable of enhancing AChR clustering[97] and proved to be successful in the treatment for severe AChR deficiency syndromes[98–100,111], but a strong molecular explanation for the observed favorable effects was still missing. This study provides possible molecular explanations for the reported successful use of such treatments by relating CMS phenotypic severity with formation of AChR clusters at the motor neuron membrane. Several of the genes identified in this analysis do not have previous associations with the

NMJ, such as the Usher syndrome and Retinitis pigmentosa associated gene; USH2a, identified as a copy number loss in patient 2. Previous studies have commented on USH2A presence on the basement membranes of perineurium nerve fibers[112,113], however, further studies in a mammalian model and/or using zebrafish mutants rather than transient knockdown will be required to determine the presence of USH2a at the NMJ, and whether loss of USH2a alone can impact NMJ signaling or whether co-occurrence with *CHRNE* CMS is required. In this regard, we report evidence of USH2A presence at the tibialis anterior muscle (Supplementary Fig. 9A) and the soleus muscle (Supplementary Fig. 9B) (Methods) in 10-week-old C57BL/6J (Jax) male mice. Additional functional work is also required to ascertain the importance of other potential modifiers identified in this study. Particularly, a prospective analysis on the potential NMJ involvement of the unique variants detected for the non-severe group could be of special interest for the study of CMS, potentially discerning their functional relationship to causal CMS genes.

Our work represents a thorough study of a narrow population showing a differential accumulation of damaging mutations in patients with CMS who have varying phenotypic severities, building on the initial impact of *CHRNE* mutations on the NMJ. It is important to remark that CMS is of particular interest among rare diseases, since drugs that influence neuromuscular transmission can produce clear improvements in the affected patients[114]. In this sense, identifying meaningful molecular relationships between gene variants allow us to gain insight into the disease mechanisms through a biomedical multilayer network framework, paving the way for a whole new set of computational approximations for rare disease research.

## Methods

### Ethics approval
This study was approved by the Ethics committee of Sofia Medical University (protocol 4/15-April-2013). Written informed consent was obtained from all the participants in the study, including more than two indirect identifiers. The study abides by the Declaration of Helsinki; no compensation was given to the participants. All animal experiments were approved by the University of Ottawa animal care and veterinary service department (protocol #3089) and complied with the guidelines of the Canadian Council on Animal Care and the Animals for Research Act. Reporting of animal sex, age and strain details comply with the ARRIVE guidelines.

### WGS and RNA-seq
Whole genome sequencing (WGS) data have been obtained from blood using the Illumina TruSeq PCR-free library preparation kit. Sample sequencing was performed with the HiSeqX sequencing platform (HiseqX v1 or v2 SBS kit, 2 × 150 cycles), with an average mean depth coverage ≥30X. Samples have been analyzed using the RD-Connect pipeline: BWA-mem for alignment; Picard for duplicate marking and GATK 3.6.0 for variant calling. RNA sequencing (RNA-seq) data have been obtained from fibroblasts, using Illumina TruSeq RNA Library Preparation Kit v2, sequencing with an average of 60 M reads per sample (paired-end 2 × 125 cycles). Data has been processed with the following pipeline[115]: STAR 2.35a for alignment, RSEM 1.3.0 for quantification, and GATK 3.6.0 for variant calling. All analyses have been performed using the human genome GRCh37d5 as reference.

### Copy number variations
Copy Number Variations (CNVs) have been extracted using ClinCNV (https://github.com/imgag/ClinCNV) by employing a set of Eastern European samples as a background control group. Out of the 569 autosomal CNVs we selected as potential candidates the CNVs of the following types that overlapped with protein-coding genes: 1) whole gene gains or losses, and 2) partial losses (deletions overlapping with exons but not with the whole gene). The list of potential candidates included 55 CNVs that created a total of 82 whole gene gains or losses and 28 partial losses.

### Compound heterozygous variants
Compound heterozygous variants have been obtained by phasing the WGS variant calls with the RNA-seq aligned BAM files using phASER[116]. At first, variants are imputed using Sanger Imputation Service with EAGLE2 pre-phasing step[117]. PhASER is then applied to extend phased regions to gene-wide haplotypes. By accurately reflecting the muscle transcriptome, fibroblasts have been previously proved to be excellent and minimally invasive diagnostic tools for rare neuromuscular diseases[118]. We then annotated variants with eDiVA tool (www.ediva.crg.es)[119], and removed all mutations with Genome Aggregation Database (gnomAD)[120] that show allele frequency >3% globally, all variants outside exonic and splicing regions using Ensembl annotation, all synonymous mutations, and all variants with read depth (coverage) smaller than 8. Afterwards we selected all genes with at least two hits on different alleles as affected by damaging compound heterozygous variants. Each sample has been processed individually throughout the whole process.

### Monolayer community detection
We performed a network community detection analysis using the Louvain clustering algorithm[121] implemented in R package igraph (https://igraph.org/) with default parameters. We carried out the analysis using three (monolayer) networks, obtained from Reactome database[32], from the Recon3D Virtual Metabolic Human database[33] (both downloaded in May 2018), and from the Integrated Interaction Database (IID)[34] (downloaded in October 2018). Additional information on network connectivity metrics (e.g. node centrality distributions and specific centrality information for severe-specific module genes) is conveniently provided as a Jupyter Notebook, accessible at the following link: https://github.com/ikernunezca/CMS/blob/master/Scripts/Multilayer_Network_Information_and_Connectivity_Patterns.ipynb.

All gene identifiers of each network were converted to NCBI Entrez gene identifiers using R packages AnnotationDbi v1.44.0 and org.Hs.eg.db v3.7.0 (http://bioconductor.org/). After detecting the community structure from each layer independently, we retrieved the community membership of the genes of interest, henceforth called CMS linked genes, i.e. known CMS causal genes, and severe and not-severe compound heterozygous variants and CNVs. We then defined a community similarity measure as Jaccard Index, i.e. the number of shared genes of interest between the communities divided by the sum of the total number of genes of each community.

### Multilayer community detection
We constructed a multilayer gene network composed of the three monolayer networks described in the previous section (Reactome, Virtual Metabolic Human and Integrated Interaction Database). Each of these three networks represents one layer of the multilayer network and, in general, three facets of fundamental molecular processes in the cell (Suppl. Figure 10). The multilayer community detection analysis was performed using MolTi software[19], which adapts the Louvain clustering algorithm with modularity maximization to multilayer networks. The algorithm is parametrized by the resolution ($\gamma$): the higher the value of $\gamma$, the smaller the size of the detected multilayer communities. By varying the resolution parameter $\gamma$ it is possible to uncover the modular structure of network communities[122].

By exploring a wide range of resolution parameter values, we identified $\gamma = 4$ (727 communities, each one composed of 26.46 genes on average) as an extreme value before both size and number of the detected multilayer communities stabilize (Supplementary Fig. 11). The most dramatic changes in number and composition of detected communities are observed in the resolution parameter interval $\gamma \in$

(0,4]. We, therefore, used this parameter interval to test the hypothesis that disease-related genes consistently appear in the same multilayer communities, as well as to identify modules containing CMS linked genes within them. In this analysis, we define a module as a group of CMS linked genes that are systematically found to be part of the same multilayer community while increasing the resolution parameter (see Supplementary Information, Multilayer community detection analysis).

## Additional analyses

We retrieved known CMS causal genes from the GeneTable of Neuromuscular Disorders (http://www.musclegenetable.fr, version November 2018)[123]. Segregation analysis of WGS data has been performed using Rbbt[124]. DisGeNET database[37] was downloaded in November 2018. The association between CMS severity, demographic factors and clinical tests was assessed with a two-tailed Fisher's test using R statistical environment (www.R-project.org). Networks were rendered with Cytoscape[125]. We used VCFtools[126] to compute familial relatedness $\Omega$ among patients, scaled to $-\log_2(2\Omega)$. We used Enrichr[127] for the functional enrichment analysis of the gene lists under study. We used Ensembl Variant Effect Predictor (VEP)[78] to assess the impact of the compound heterozygous variants in the genes of the severe-specific largest module. Expression levels in tissues of interest (GTEx and Illumina Body Map) were retrieved from EBI Expression Atlas (www.ebi.ac.uk/) by filtering with the following keywords: 'nerve', 'muscle cell', 'fibroblast' and 'nervous system' (0.5 TPM default cutoff). We used Expression Atlas expression level categories: low (0.5 to 10 TPM), medium (11 to 1000 TPM), and high (more than 1000 TPM). Synaptic localization was retrieved from the UniProt database (https://www.uniprot.org/).

## Western Blot and Immunostaining of USH2A on mouse neuromuscular junctions

For Western Blotting 40 mg of protein was run on a 10% gel and transferred to a membrane using the BioRad Trans Turbo semi-dry transfer machine. The membrane was blocked in milk for 1 h and Usherin (FabGennix, USH2A-112AP, 1:2000) was added (5% BSA in TBST) overnight. Secondary antibodies were diluted 1:1000 in milk.

Labeling of the neuromuscular junction (NMJ) was performed on soleus muscle in 10-week-old C57BL/6 J (Jax) male mice. Muscles were washed in ice-cold PBS (2 × 10 min) and then separated out into small bundles under a stereo-microscope. They were fixed overnight at 4 °C in 2% PFA, washed 2 × 1 h with ice-cold PBS, and treated with Analar Ethanol and Methanol both at −20 °C (10 min each). Tissues were then incubated with blocking/permeabilization solution (5% horse serum, 5% BSA, 2% Triton X-100 in PBS) for 4 h (room temp (RT)) with gentle agitation.

Muscle bundles were incubated with antibodies, diluted in blocking buffer without triton, against Usherin-FITC (Rb polyclonal, FabGennix USH.101-FITC,1:100) overnight (4 °C) with agitation and then for a further 2 h (RT) the following morning. Muscles were then washed in blocking buffer 4 × 1 h (RT) and incubated with Alexa 594-Conjugated α-Bungarotoxin (ThermoFisherScientific, B13423, 1:250), for 4 h (RT). Samples were washed 4 × 1 h in PBS and then mounted using Vectashield hardset mounting medium. Images were captured using Olympus FV1000c scanning confocal microscope using FV1000 application software (FV10-ASW) software at x63 oil immersion objective.

Animals were housed under 12 h light/dark cycles and had ad libitum access to standard chow (Teklad Global 18% protein Rodent Diet) and water.

## Reporting summary

Further information on research design is available in the Nature Portfolio Reporting Summary linked to this article.

## Data availability

WGS metadata and variant data, and patient phenotypic descriptions have been deposited in the RD-Connect GPAP: https://platform.rd-connect.eu/#/. This data is available under controlled access for registered users of the GPAP. Details on access to GPAP can be found in: https://platform.rd-connect.eu/userregistration. Biobank sample accession identifiers are provided in Supplementary Table 1. The raw RNA-Seq dataset analyzed in this study is not publicly available due to sensible content (patient molecular data on a rare disease). Minimal, pre-processed RNA-Seq data for reproducibility is provided within the github repository of the project: https://github.com/ikernunezca/CMS/tree/master/data/fibroblast_expression. Given the sensitive nature of this data, accessibility should be requested to Hanns Lochmüller (Children's Hospital of Eastern Ontario Research Institute; Ottawa, Canada) at hlochmuller@toh.ca. A reasonable timeframe for a response could be within two weeks of sending the request. All source data and code files for reproducing all the Figures are provided within the github repository of the project: https://github.com/ikernunezca/CMS. Information on the source data can be accessed from: https://github.com/ikernunezca/CMS/blob/master/Source_Information_README. We specifically provide the Cytoscape Session file ('cys') containing all the plots used to produce Figs. 3 and 4, as well as Supplementary Figs. 3, 6, and 8 in this link: https://github.com/ikernunezca/CMS/blob/master/Cytoscape_Session/CMS_Session.cys. Specific input Source Data files for creating the Cytoscape Session used to build Figs. 3, 4A, B, 6 and 8 can be accessed from the following link as csv files: https://github.com/ikernunezca/CMS/tree/master/Cytoscape_Session. Additionally, the Cytoscape Session provides an extra plot with the incident interactions considered to render Fig. 5. Supplementary Fig. 1 source data is provided as Supplementary Table 1. Input Data for reproducing Supplementary Fig. 2 can be accessed from: https://github.com/ikernunezca/CMS/tree/master/data/InputGenes. Input for plotting Supplementary Figure 11 as well as information on the files is available at: https://github.com/ikernunezca/CMS/tree/master/data/MolTi/Community_Analysis.

## Code availability

All generetad code and the Cytoscape session rendering Figs. 3 and 4, as well as Supplementary Figs. 3, 6 and 9 are available for reproducibility purposes at: https://github.com/ikernunezca/CMS. The analysis of multilayer communities can also be performed using CmmD[110] (https://github.com/ikernunezca/CmmD) with parameters: resolution_start: 0, resolution_end: 4, interval: 0.5 and the CMS linked genes as nodelist. Code can also be referenced using Zenodo[128].

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

## Acknowledgements

The authors acknowledge the donors and families, Daniel Rico (Newcastle University) for his contribution in early stages of the project, Anaïs Baudot (Aix Marseille Université and Barcelona Supercomputing Center) for her careful revision of the manuscript, Miguel Vázquez (Barcelona Supercomputing Center) for advising about Rbbt analysis, Jon Sánchez-Valle (Barcelona Supercomputing Center) and Núria Olvera (Barcelona Supercomputing Center and IDIBAPS) for the insightful discussions. The authors also acknowledge the corresponding funding institutions: The NeurOmics and RD-Connect projects have been funded by the European Union's Seventh Framework Programme for research, technological development and demonstration under grant agreements no 2012-305121 and 2012-305444. I.N.C. was supported by a grant for pre-doctoral contracts for the training of doctors (Project ID: SEV-2015-0493-18-2) (Grant ID: PRE2018-083662) from the Spanish Ministry for Science, Innovation and Universities, and the EU project EVENFLOW under Horizon Europe agreement No. 101070430. E.O. was supported by an AFM-Téléthon postdoctoral fellowship for the duration of this work. H.L. receives support from the Canadian Institutes of Health Research (Foundation Grant FDN-167281), the Canadian Institutes of Health Research and Muscular Dystrophy Canada (Network Catalyst Grant for NMD4C), the Canada Foundation for Innovation (CFI-JELF 38412), and the Canada Research Chairs program (Canada

Research Chair in Neuromuscular Genomics and Health, 950-232279). V.G. was a research fellow of the Alexander von Humboldt Foundation. D.C. was supported by the European Commission's Horizon 2020 Program, H2020-SC1-DTH-2018- 1, iPC - individualizedPaediatricCure (ref. 826121).

## Author contributions

T.C., I.T. and V.G. collected and processed the biopsies; H.L. and R.T. coordinated data sharing; A.T., P.A.C.T., S.B. and S.C. coordinated and performed the omics data analysis with Y.A., S.L., M.R. and M.B.; E.O. and S.S. performed the experimental validations; D.C. and A.V. coordinated the multilayer network analysis performed by I.N.C. All authors contributed to the writing and revising of the manuscript.

## Competing interests

The authors declare no competing interests.

## Additional information

[1]Barcelona Supercomputing Center (BSC), Plaça Eusebi Güell, 1-3, 08034 Barcelona, Spain. [2]MRC London Institute of Medical Sciences, Du Cane Road, London W12 0NN, UK. [3]Institute of Clinical Sciences, Faculty of Medicine, Imperial College London, Hammersmith Hospital Campus, Du Cane Road, London W12 0NN, UK. [4]Coordination Unit Spanish National Bioinformatics Institute (INB/ELIXIR-ES), Barcelona Supercomputing Center, Barcelona, Spain. [5]Children's Hospital of Eastern Ontario Research Institute, Ottawa, ON, Canada. [6]Brain and Mind Research Institute, University of Ottawa, Ottawa, ON, Canada. [7]Department of Human Genetics, Yokohama City University Graduate School of Medicine, Yokohama, Japan. [8]Department of Pediatrics, Aichi Medical University, Nagakute, Japan. [9]John Walton Muscular Dystrophy Research Centre, Translational and Clinical Research Institute, Newcastle University, Newcastle upon Tyne, United Kingdom. [10]Newcastle Hospitals NHS Foundation Trust, Newcastle upon Tyne, United Kingdom. [11]Center for Molecular and Biomolecular Informatics, Radboud Institute for Molecular Life Sciences, Radboud university medical center, Nijmegen, The Netherlands. [12]Department of Neurology, Expert Centre for Hereditary Neurologic and Metabolic Disorders, Alexandrovska University Hospital, Medical University-Sofia, Sofia, Bulgaria. [13]Department of Cognitive Science and Psychology, New Bulgarian University, Sofia 1618, Bulgaria. [14]Clinic of Neurology, University Hospital Sofiamed, Sofia University St. Kliment Ohridski, Sofia, Bulgaria. [15]Centro Nacional de Análisis Genómico (CNAG-CRG), Center for Genomic Regulation, Barcelona Institute of Science and Technology (BIST), Barcelona, Catalonia, Spain. [16]Universitat Pompeu Fabra (UPF), Barcelona, Spain. [17]Departament de Genètica, Microbiologia i Estadística, Facultat de Biologia, Universitat de Barcelona (UB), Barcelona, Spain. [18]Division of Neurology, Department of Medicine, The Ottawa Hospital, Ottawa, ON, Canada. [19]Department of Neuropediatrics and Muscle Disorders, Medical Center – University of Freiburg, Faculty of Medicine, Freiburg, Germany. [20]ICREA, Pg. Lluís Companys 23, 08010 Barcelona, Spain. ✉e-mail: davide.cirillo@bsc.es

