## [Peer Review file · Nature Communications]

REVIEWER COMMENTS

Reviewer #1 (Remarks to the Author):

This is essentially a data mining exercise trying to address an important but largely unanswered question for many genetic disorders, which is why do many cases with apparently the same mutation have a very different phenotype/severity.

The authors utilise genetic disorders of the neuromuscular junction as their model system, and use the common mutation in CHRNE present in a patient cohort that is relatively isolated population in genetic terms. The cohort is divided largely into severe and mild phenotypes and compare these groups for multiple level data sets. They propose from their data that variations (largely single nucleotide compound variants) in extracellular matrix proteins are likely to be responsible for the differing patient severity.

The drawback of the work is a lack of specifics. I found it difficult to decipher what actually constituted a severe patient and how this was determined.

Although a number of variants in the extracellular matrix proteins were proposed as deleterious there were no actual experiments to demonstrate actual functional effects, rather just predictions of deleterious effects. The authors did use a zebrafish model to propose a role for USH2A at the neuromuscular junction. However, given that they are looking at a human disease phenotype a zebrafish cannot be taken as reliable. It would have been important to show the presence of this protein at the neuromuscular junction in preferably human, but at least in mouse neuromuscular junctions. It is important to note that the neuromuscular junction is located at only a very small region of a muscle fiber, and so detection of a protein in muscle is not necessarily a good indicator of influence on neuromuscular junction function. To be convinced of a true functional role for the other proposed extracellular matrix proteins and Agrin in the disease severity I would suggest functional analysis is done to show some influence on how the neuromuscular junction functions. In their analysis, do the authors take into account the large size and relative lack of conservation in their code.

The authors could perhaps elaborate more fully on the role of CHRNE and why mutations in this gene are the most frequent cause of congenital myasthenia and how this causes the characteristic features of disorder. The description in the supplementary data for CHNE disease should be expanded and could be included in the main text.

Reviewer #2 (Remarks to the Author):

The manuscript by Núñez-Carpintero and colleagues aims to address the important challenge of understanding differences in disease severity among rare disease patients with the same causative genetic mutation. To investigate the role of additional genomic variants in individual patients they propose a framework based on a multilayer network that combines protein-protein, biological pathway, and metabolomic interactions. They apply their framework to a well characterized cohort of patients displaying mild and severe versions of Congenital Myasthenic Syndromes (CMS). The analyses suggest that variants affecting the extracellular matrix as well as certain receptors may play an important role for disease severity. Finally, the authors use a zebra fish model to evaluate the potential involvement of a gene with no known link to neuromuscular junction disorders.

The presented work is conceptually and methodologically interesting. It showcases a systematic framework for using complementary information to understand differences in rare disease severity. The framework is potentially applicable for other rare diseases and could also be generalized for more common, complex phenotypes, thus having broader implications for personalized medicine.

The manuscript contains the full workflow from individual variant interpretation, to network integration, to hypothesis generating of a biological mechanism with potential therapeutic implications, and finally experimental validation of a new candidate gene. While not all results of this workflow are equally strong, I do appreciate their comprehensive nature as a whole.

The manuscript is well written and easy to follow also for non experts in NMJ disorders, like myself, thanks to a comprehensive introduction and thorough discussion of individual variants and their potential role in disease severity. Note, however, that I cannot judge in detail how plausible the suggested mechanisms are. Similarly, I cannot judge how convincing the employed zebra fish model is for NMJ disorders.

An area where I see potential for improvement is the presentation of the multilayer network and respective analyses:

The manuscript contains little too no information about the individual network layers, nor about their combination or how they compare to each other. How many nodes and edges are there? Are the layers comparable in terms of size and coverage? Do they have different overall connectivity patterns? To which extent are the different layers complementary? I think it would be important to add more information in this direction to help the reader understand the basic data that the work is built upon.

Along the same lines, it would be interesting to see the different interaction types displayed in Figures 3 and 4 to better appreciate the usefulness of the multilayer approach. Do the three layers contribute equally to the final modules?

Finally, a more conceptual question occurred to me while going through the manuscript: Do the authors consider mild disease as simply the absence of additional variants leading to a more severe outcome? Or would they expect that the unique variants identified in the mild cases may have protective effects? I understand that this question will likely have no general answer, but since the basic hypothesis of how variants modulate disease outcome would impact both the methodology for searching for disease modifying variants, as well as their interpretation, I believe a brief discussion would be instructive.

Reviewer #3 (Remarks to the Author):

Congenital myasthenia syndromes are a diverse group of neuromuscular junction disorders. CMS phenotypic severity varies dramatically, likely because they are caused by different mutations of same and/or different genes. Earlier studies were hampered by relatively small populations of CMS patients/samples. This multi-center study was able to group patients into severe disease group and no-sever group. Analysis of three main types of genetic variations (SNPs, CNVs, and compound heterozygous variants) did not render any SNPs as could be considered a unique cause of disease severity. The authors developed an algorithm to compare patient genomic data with the information provided by a biomedical knowledge multilayer network and were able to reveal functional relationships of new genetic modifiers responsible for different phenotypic severity levels. They explored phenotypes of knocking down one such modifier USH2A in zebrafish and demonstrated that the strategy seems to work.

The paper reports on a new workflow or strategy to explain why some patients with same mutation vary in disease severity, which could be applied to other rare disorders. I have only minor suggestions. First, the literature citation should be more accurate. For example, when the agrin-Lrp4-Musk pathway was described, a paper by Whicher, Philbin, and Aronson (2018) was cited. The paper was NOT on the pathway or NMJ at best; it was entitled "An Overview of the Impact of Rare Disease Characteristics on Research Methodology." Recent reviews on NMJ formation/signaling should cited including Li et al., Ann Rev Physiology 2018. Second, the paper contains many typos. For example, line 54, This work showcase how... It should read "showcases". Extensive proof reading is needed.

POINT-TO-POINT RESPONSE TO THE REVIEWERS

Reviewer #1 (Remarks to the Author):

1) This is essentially a data mining exercise trying to address an important but largely unanswered question for many genetic disorders, which is why do many cases with apparently the same mutation have a very different phenotype/severity. The authors utilise genetic disorders of the neuromuscular junction as their model system, and use the common mutation in CHRNE present in a patient cohort that is relatively isolated population in genetic terms. The cohort is divided largely into severe and mild phenotypes and compare these groups for multiple level data sets. They propose from their data that variations (largely single nucleotide compound variants) in extracellular matrix proteins are likely to be responsible for the differing patient severity.

We thank the reviewer for their appreciation of our work.

2) The drawback of the work is a lack of specifics. I found it difficult to decipher what actually constituted a severe patient and how this was determined.

Determination of phenotypes was performed by expert clinicians (I.T. and V.G.), taking a detailed medical history and performing a full neurological exam. Specifically, severe CMS patients present severe generalised muscle fatigue, impaired myopathic gait, hyperlordosis and reduced FVC (forced vital capacity). For better clarity, we have added this information to the manuscript, on top of the already observed association between FVC clinical trait and severity in our data: (**Lines 115-120:** “Clinically, CMS severity ranges from minor symptoms (e.g., exercise intolerance) to more severe CMS forms and is dependent on the causal genetic impairments (Abicht et al., 1993; Della Marina et al., 2020). Severe CMS is typically presented with reduced Forced Vital Capacity (FVC), severe generalized muscle fatigue and weakness, proximal and bulbar muscle fatigue and weakness, impaired myopathic gait and hyperlordosis.” and **Lines 122-126:** “Out of the tested demographic factors (age, sex) and clinical tests (speech, mobility, respiratory dysfunctions, among others) FVC and shoulder lifting ability show a significant association with the severity classes (two-tailed Fisher’s exact test p-values of 0.0128 and 0.0418, respectively; **Suppl. Figure 1.**”).

Such clinical characteristics were originally collected using a Congenital Myasthenic Syndrome Combined Child/Adult Form designed in the context of the European project that supported the study. This form is adapted from the QMG (Quantitative Myasthenia Gravis Scale), which is widely used in clinical trials for Myasthenia Gravis, an autoimmune disease of the neuromuscular junction exhibiting similar symptoms and impairments as CMS. The adapted form has also been used in children with CMS (Della Marina et al. *Front Hum Neurosci.* 2020) (PMID 33364925).

3) Although a number of variants in the extracellular matrix proteins were proposed as deleterious there were no actual experiments to demonstrate actual functional effects, rather just predictions of deleterious effects.

We thank the reviewer for this remark. We are aware of the limitations of the available resources for deleterious effect predictions of variants. For this reason, and to ensure clarity, we consistently refer to them as “predicted to be deleterious” throughout the text (**‘Functional consequences of variants in the severe-specific module’**). Also, we provide extensive supporting evidence of possible (mis)functional implications of such variants, including a curation of synaptic localization (**Table 2, ‘Synaptic localization’ and ‘Localization (UniProt)’**) and of the literature reporting either functional involvement at the NMJ level or overlapping phenotypes caused by genetic impairments of such genes (particularly, myopathies and muscular dystrophies) (**Section: ‘Personalized analysis of the severe cases’**). Finer description of the information provided by this literature is given in the response to comment no. 5 from the reviewer. Additionally, provided that previous literature relating *USH2A* to potential functional involvements at the NMJ was not available, we performed knockdown experiments on the orthologous gene from zebrafish, *ush2a*. Moreover, we provide initial results of *USH2A* colocalization experiments performed on 10 week old mice in the response to comment no. 4 from the reviewer.

4) The authors did use a zebrafish model to propose a role for *USH2A* at the neuromuscular junction. However, given that they are looking at a human disease phenotype a zebrafish cannot be taken as reliable. It would have been important to show the presence of this protein at the neuromuscular junction in preferably human, but at least in mouse neuromuscular junctions.

Zebrafish has been widely used in previous literature to model human neuromuscular diseases. A recent comprehensive review on zebrafish modelling of neuromuscular disorders such as Amyotrophic lateral sclerosis (ALS), Charcot-Marie-Tooth Disease and Myasthenia gravis, is that of Singh et al. *Front Mol Neurosci.* (2022) (PMID: 36583079). For the particular case of CMS, Müller et al. *Hum Mol Genet.* (2010) (PMID: 20147321) demonstrated the role of *DOK7* (a causal CMS gene) on the development of the NMJ structure using a zebrafish model. Senderek et al. *Am J Hum Genet.* (2012) (PMID: 21310273) shown how *GFPT-1* mutations in zebrafish recapitulate human CMS pathology using morpholino (MO)-based strategies to knockdown the zebrafish orthologue (which is the same strategy provided in our experiments). Tian et al. *Cell. Mol. Life Sci.* (2019) (PMID: 30327840) studied the potential pathways involved in the phenotype driven by *LPR4* mutations using a morpholino-based zebrafish model.

Furthermore, the recent publication by co-author Emily O’Connor (who performed the presented *us2ha* zebrafish experiments) provided important analysis on the role of *AGRN* in *MYO9A* CMS MO-knockdown zebrafish models (O’Connor et al. *Cells.* 2019) (PMID: 31394789). Previously, the *MYO9A* model was used to identify CMS causal variants (O’Connor et al. *Brain.* 2016) (PMID: 27259756).

Still, we do understand the concern of the reviewer on the reliability of zebrafish as a model for a human disease. Accordingly, here we present initial results of USH2A tissue western blots performed on 10 week old mice (**see below**). Using a specific, monoclonal antibody we detect a single band of the expected size in the spinal cord, liver and muscle (tibialis anterior) which is compatible with the ubiquitous expression of the protein in multiple tissues. Using immunofluorescence labelling we were able to detect USH2A (Rb polyclonal Anti-Usherin-FITC), FabGennix) in mouse brain and at some neuromuscular junctions (α -Bungarotoxin) in muscle (**see below**).

5) It is important to note that the neuromuscular junction is located at only a very small region of a muscle fiber, and so detection of a protein in muscle is not necessarily a good indicator of influence on neuromuscular junction function. To be convinced of a true functional role for the other proposed extracellular matrix proteins and Agrin in the disease severity I would suggest functional analysis is done to show some influence on how the neuromuscular junction functions.

For the particular case of agrin (**AGRN**), the recent publication from Jacquier et al. (2022) (PMID: 35948834), where manuscript co-authors Emily O'Connor and Hanns Lochmüller are involved, provides extensive functional insights on the functional involvement of variants

affecting agrin secretion to the NMJ in severe CMS cases. Retention of agrin at the endoplasmic reticulum significantly reduces the formation of AChR clusters and favours neuronal apoptosis. We also provide here curated literature on the functional roles of the proposed candidate proteins. This citations have also been added to the main manuscript ('Personalized analysis of the severe cases' and 'Functional consequences of variants in the severe-specific module')

-TNXB: *Tnxb*^{-/-} mice suffer of myopathic phenotypes (Matsumoto and Aoki, 2020) (PMID: 33335533); (Okuda-Ashitaka and Matsumoto, 2023) (PMID: 37007968)

-TNC: Tenascin-c mediates acute ECM damage repair. (Flück et al. 2008) (PMID: 18757758) (Sorensen et al. 2018) (PMID: 29466693)

-VCAN: Versican expression is significantly increased in dystrophic muscles, and has been recently suggested as a potential target for muscular dystrophy amelioration (McRae et al., 2017, 2020) (PMID: 29211034, PMID: 32632164).

-LAMA2: *LAMA2* mutations reduce neuromuscular junction folds in animal models, with a correlation of the protein absence with the severity of the phenotype (Rogers and Nishimune, 2017) (PMID: 27614294). Bi-allelic loss of function mutations in *LAMA2* cause a form of congenital muscular dystrophy with peripheral nerve involvement in humans (OMIM 156225).

-COL15A1: Lack of this collagen type leads to skeletal muscle myopathy phenotypes in mutated mice (Eklund et al. 2001) (PMID: 11158616) and zebrafish models (Guillon et al. 2016) (PMID: 26937007), where it is key in guiding motor axon development.

-CHGB: Gros-Louis et al. (2009) reported the potential involvement of *CHGB* rs742710 variant (the same as patient 12) in ALS. Importantly, this publication reports on how this variant causes *CHGB* protein granules to stay stuck within the endoplasmic reticulum / transGolgi network.

-HSPG2: Perlecan (*HSPG2* encoded protein) is present on skeletal muscle basal lamina (Larrain et al. 1997) (PMID: 9260911) (Carmen et al. 2019) (PMID: 29924302). The role of Perlecan in skeletal muscle development has also been studied using zebrafish models (Zoeller et al. 2008) (PMID: 18426981). Bi-allelic loss of function mutations in *HSPG2* cause a form of Schwartz-Jampel syndrome that often presents with fatigable weakness (OMIM 142461)

-LOXL3: Involved in myofiber extracellular matrix development by improving integrin signalling through fibronectin oxidation and interaction with laminins in mice models. *LOXL3* mutant mice myofibers present downregulated integrin signalling, and aberrant fibronectin matrix formation (Kraft-Sheleg et al., 2016) (PMID: 26954549).

Curated literature on skeletal muscle involvement for *USH2A* is non-existent, which is the main reason it was targeted for the presented experimental validation. Human Protein Atlas RNA-seq data indicates remarkable signals from brain excitatory neurons. Limited expression is also observed for some fibroblast clusters of Skeletal Muscle (<https://www.proteinatlas.org/ENSG00000042781-USH2A/single+cell+type/skeletal+muscle>). We provide evidence that *USHA2* can be detected at the mouse NMJ co-localizing with the AChR by immunohistochemistry.

6) In their analysis, do the authors take into account the large size and relative lack of conservation in their code.

We thank the reviewer for this remark. All scripts have been now reworked (with a particular focus on computational efficiency) into Jupyter Notebooks for easier understanding of the presented computational analysis. The new versions of these scripts are now available in Github (<https://github.com/ikernunezca/CMS>). We agree that there was a need for a better presentation of these scripts, and therefore we thank the reviewer for pointing out this issue.

7) The authors could perhaps elaborate more fully on the role of *CHRNE* and why mutations in this gene are the most frequent cause of congenital myasthenia and how this causes the characteristic features of disorder. The description in the supplementary data for *CHRNE* disease should be expanded and could be included in the main text.

Following reviewer's suggestion, we have improved the description of *CHRNE*'s role as AChR subunit in the supplementary data (**Lines 143-148: *CHRNE***, which encodes the ϵ subunit of the AChR receptor, accounts as causative for ~50% of all reported CMS cases, although frequencies might vary depending on ethnicity (Abicht et al., 1993; Finsterer, 2019). The high prevalence of ϵ subunit mutations may be the result of partial compensation of its functionality by the embryonic γ (encoded by *CHRNG*), which is substituted after birth given its lower conductance levels. Mutations in other subunits reduce patient survival as no compensation mechanism occurs (Engel et al., 1996).”)

Although we agree that the whole section could be added to the main text, we currently opted out of doing so given the already considerable length of the manuscript, as we believe that, given the scope of our work, a major focus should be given to the multilayer network approach and the personalised analysis of the patient impairments.

Reviewer #2 (Remarks to the Author):

The manuscript by Núñez-Carpintero and colleagues aims to address the important challenge of understanding differences in disease severity among rare disease patients with the same causative genetic mutation. To investigate the role of additional genomic variants in individual patients they propose a framework based on a multilayer network that combines protein-protein, biological pathway, and metabolomic interactions. They apply their framework to a well characterized cohort of patients displaying mild and severe versions of Congenital Myasthenic Syndromes (CMS). The analyses suggest that variants affecting the extracellular matrix as well as certain receptors may play an important role for disease severity. Finally, the authors use a zebra fish model to evaluate the potential involvement of a gene with no known link to neuromuscular junction disorders.

The presented work is conceptually and methodologically interesting. It showcases a systematic framework for using complementary information to understand differences in rare disease severity. The framework is potentially applicable for other rare diseases and could also be generalized for more common, complex phenotypes, thus having broader implications for personalized medicine.

The manuscript contains the full workflow from individual variant interpretation, to network integration, to hypothesis generating of a biological mechanism with potential therapeutic implications, and finally experimental validation of a new candidate gene. While not all results of this workflow are equally strong, I do appreciate their comprehensive nature as a whole.

We thank the reviewer for their appreciation of our work.

1) The manuscript is well written and easy to follow also for non experts in NMJ disorders, like myself, thanks to a comprehensive introduction and thorough discussion of individual variants and their potential role in disease severity. Note, however, that I cannot judge in detail how plausible the suggested mechanisms are. Similarly, I cannot judge how convincing the employed zebra fish model is for NMJ disorders.

Zebrafish has been widely used in previous literature to model human neuromuscular diseases. A recent comprehensive review on zebrafish modelling of neuromuscular disorders such as Amyotrophic lateral sclerosis (ALS), Charcot-Marie-Tooth Disease and Myasthenia gravis, is that of Singh et al. *Front Mol Neurosci.* (2022) (PMID: 36583079). For the particular case of CMS, Müller et al. *Hum Mol Genet.* (2010) (PMID: 20147321) demonstrated the role of *DOK7* (a causal CMS gene) on the development of the NMJ structure using a zebrafish model. Senderek et al. *Am J Hum Genet.* (2012) (PMID: 21310273) shown how *GFPT-1* mutations in zebrafish recapitulate human CMS pathology using morpholino (MO)-based strategies to knockdown the zebrafish orthologue (which is the same strategy provided in our experiments). Tian et al. *Cell. Mol. Life Sci.* (2019) (PMID: 30327840) studied the potential pathways involved in the phenotype driven by *LPR4* mutations using a morpholino-based zebrafish model.

Furthermore, the recent publication by co-author Emily O'Connor (who performed the presented *us2ha* zebrafish experiments) provided important analysis on the role of *AGRN* in *MYO9A* CMS MO-knockdown zebrafish models (O'Connor et al. *Cells*. 2019) (PMID: 31394789). Previously, the *MYO9A* model was used to identify CMS causal variants (O'Connor et al. *Brain*. 2016) (PMID: 27259756).

Still, we do understand the concern of the reviewer on the reliability of zebrafish as a model for a human disease. Accordingly, here we present initial results of USH2A tissue western blots performed on 10 week old mice (see below). Using a specific, monoclonal antibody we detect a single band of the expected size in spinal cord, liver and muscle (tibialis anterior) which is compatible with the ubiquitous expression of the Usha2 protein in multiple tissues. Using immunofluorescence labelling we were able to detect USH2A (Rb polyclonal Anti-Usherin-FITC), FabGennix) in mouse brain and at some neuromuscular junctions (α -Bungarotoxin) in muscle (see below).

An area where I see potential for improvement is the presentation of the multilayer network and respective analyses:

We agree with the reviewer on the need of a more thorough description of the particularities of the multilayer network used for the analysis. We present a new Jupyter Notebook, available in Github, focused on such particularities that can be accessed via the following link:

https://github.com/ikernunezca/CMS/blob/master/Scripts/Multilayer_Network_Information_and_Connectivity_Patterns.ipynb

We discuss the particular characteristics of the multilayer network observed through the presented analysis in the responses to the following reviewer's comments.

2) The manuscript contains little too no information about the individual network layers, nor about their combination or how they compare to each other. How many nodes and edges are there? Are the layers comparable in terms of size and coverage? I think it would be important to add more information in this direction to help the reader understand the basic data that the work is built upon.

We thank the reviewer for this comment. We initially provided this information within the Methods section of the manuscript '**Monolayer community detection**'. We have moved it to the main text following reviewer's suggestion (**Lines 212/216**: *"The first network consists of 10,618 nodes (genes) and 875,436 edges, representing shared pathways between genes. The second network consists of 1,863 nodes (genes) and 902,188 edges, representing shared reaction metabolites between genes. The third network consists of 18,018 nodes (genes) and 947,606 edges, representing aggregated protein-protein interactions from all tissues."*).

The interactome layer presents the highest numbers of nodes and edges. The pathway (Reactome) layer presents a lower, yet comparable density of edges per node. Finally, the metabolomic (Recon3D Virtual Metabolic Human) presents the lowest quantity of nodes but the highest degree density. In terms of node centrality, interactome and pathways layers do present a similar distribution of the degree density per node (**Figure 2-3**).

Reactome pathways degree distribution naturally presents higher bandwidth provided its nature (genes being part of the same pathway will be connected to all members of the same pathway, which is not the case in protein-protein interaction).

Figure 2. Protein-protein interaction layer degree distribution.

Figure 3. Reactome pathways layer degree distribution.

The metabolomic layer presents a limited number of nodes in comparison to the pathways and interactome layers, with the majority of the nodes presenting high degree numbers. Again, the definition of the network edges plays a major role in such topologies, as edges exist between two nodes if a metabolite engages with the corresponding gene products in terms of production and utilisation (**Figure 4**).

Figure 4. Metabolome layer degree distribution.

Degree distributions also translate into comparable behaviour between Reactome and interactome layers in terms of node betweenness (**Figure 5**). Behaviour of node betweenness distribution is also coherent in the case of the metabolomic layer, provided the high degree density (**Figure 6**).

Figure 5. Normalised node betweenness distributions for protein-protein interaction and reactome layers.

Figure 6. Normalised node betweenness distribution for the metabolomic layer.

For the 15 genes of the severe-specific module, we can observe different centrality behaviours depending on the layer. For the interactome layer, *TNC*, *VCAN*, *AGRN*, *HSPG2* and *PLEC* are the top 5 nodes both in terms of node degree as well as for node betweenness (**Figure 7**). For the pathways layer, *HSPG2*, *LAMA5*, *TNC*, *AGRN* and *VCAN* both in terms of node degree as well as for node betweenness (**Figure 8**).

Figure 7. Degree (left) and betweenness (right) boxplots for severe-specific module genes at the protein-protein interaction layer.

For the metabolic layer, an important observation should be highlighted. None of the 15 nodes found in the detected severe-specific module exist in the metabolomic layer, as they do not encode for enzymes or other proteins related to metabolism. Still, the importance of this layer is observed for the particular case of *ACOT2* gene, which is mutated in Patient 3.

Figure 8. Degree (left) and betweenness (right) boxplots for severe-specific module genes at the pathway interaction layer.

Indeed, while Patient 3 is the only severe individual not presenting mutations within the main detected module, its mutated gene *ACOT2* appears in the metabolomic network and presents several connections with known CMS causal genes through all the other layers. We comment this observation in the manuscript and pinpoint *ACOT2* as an interesting gene for further studies:

“Moreover, variant rs151154986 (AF=2.18E-02) affects the acyl-CoA thioesterase ACOT2, which generate CoA and free fatty acids from acyl-CoA esters in peroxisomes (Grevengoed, Klett, and Coleman 2014). A role for CoA levels in skeletal muscle for this enzyme class has been previously described (Li et al. 2015). While ACOT2 is lost early during the module detection process, community detection at the individual layer level (i.e. Louvain community detection for each network) revealed relationships with causal CMS genes throughout all layers of the multilayer network system (Supplementary Figure 3). Namely, ACOT2 shares community membership with ALG14, DPAGT1, GFPT1, GMPPB and SLC25A1A at the protein-protein interaction network; with CHAT and SLC5A7 at the pathways level, and with GMPBB, SLC25A1 and CHAT at the metabolomic layer.”

Therefore, we focused the degree analysis of the metabolomic layer on the genes appearing in the **Supplementary Figure 3 (Metabolome, severe)**: *ACOT2*, *SLC18A3*, *ATP6V0A4*, *AMPD1*, *CES5A*, *GMPBB*, *SLC25A1*, *CHIT1*, *CHAT*, *SLC28A1*, *SLC5A7* and *GFPT1* (**Figure 9**). As it can be seen, *ACOT2* presents a very high degree given that the total number of nodes within the layer is 1863. *SLC25A1* (CMS causal gene) presents the highest betweenness.

Figure 9. Degree (left) and betweenness (right) boxplots for severe-specific module genes at the metabolome interaction layer.

3) Do they have different overall connectivity patterns?

To address this question on connectivity patterns, we decided to focus on participation coefficient (PC), which is a convenient metric to calculate the extent to which a node's degree is positioned within its network community relative to other communities (Guimerá and Amaral. *Nature*. 2005) (PMID: 15729348). PC values close to 0 indicate higher intramodular importance, while values close to 1 indicate nodes whose degree / betweenness is sparse over the whole network.

For the participation coefficient within the interactome layer, *VCAN* is the top node of the module. Interestingly, *TNXB*, which is among the last nodes of the module in terms of degree and betweenness, is second overall for PC, a behaviour that is shared with *LAMA2*. *TNC*, similarly to *VCAN*, is still among the top 5 in PC, indicating the critical topological role of both genes within the module (**Figure 10**).

Figure 10. Participation coefficient (interactome layer) for the genes conforming the severe-specific module.

For the pathways layer, interesting dynamics can be observed, as *HSPG2*, which has the highest betweenness and degree of the module, presents also the highest PC, therefore presenting a considerable number of extra-module interactions. *LOXL3* presents the opposite behaviour, becoming the node with the highest PC, while presenting the lowest degree (**Figure 11**).

Figure 11. Participation coefficient (pathways interaction layer) for the genes conforming the severe-specific module.

Indeed, *LOXL3* does not present incident interactions with the other 14 CMS linked genes but it has incident interactions with other genes of the module (a total of 482 genes). We remind here, as it is clearly stated in the manuscript, that the “CMS linked genes” are those being causal for the disease and/or presenting CNVs or compound heterozygous mutations. This observation about *LOXL3*, confirmed by its PC, represents additional evidence on the ability of multilayer networks to efficiently integrate information that, otherwise, would appear disconnected. Another example of this kind is *ACOT2*, as already discussed before. For the module presenting *ACOT2* (candidate severity gene for Patient 3) (**Supplementary Figure 3**), we observe that, *GMPBB* presents the lowest PC (module formed by *ACOT2*, *GMPBB*, *SLC25A*, and *CHAT*). Of the targeted genes, *CHIT1* presents the highest PC (**Figure 12**).

Figure 12. Participation coefficient for the metabolic interaction layer.

4) To which extent are the different layers complementary?

94.39% of the edge universe (i.e. union of all network edges) is specific to individual layers. Interactome and reactome layers present the highest edge overlap (4.69% of the edge universe), followed by the overlap between reactome and metabolome (0.48%) and the overlap between interactome and metabolome layers (0.27%). Finally, only 0.17% of the edge universe is common to all three layers (**Figure 13**).

Figure 13. Edge overlap among the layers of the multilayer network. Each layer is identified by the name of the database from which the information has been obtained.

While the edge universe reveals high layer specificity, the node universe (i.e. union of all network nodes), presents much more notable overlaps. The interactome layer presents the highest amount of layer-specific nodes (a 41.75% of the node universe is formed by interactome monolayer nodes), with reactome and metabolome-specific nodes representing just a minimal fraction (2.39% and 0.13% of the node universe, respectively).

Of note, both the reactome and metabolome layers have their nodes mostly shared with other layers (95.82% and 98.71% of their layer-specific node universe, respectively). The main node pool is formed by the node overlap between the protein-protein and reactome layers (45.78% of the node universe). Finally, 9% of the node universe is common to all layers (**Figure 14**).

Figure 14. Node overlap among the layers of the multilayer network. Each layer is identified by the name of the database from which the information has been obtained.

We have now added Figures 13 and 14 of this reply to the Supplementary Information, as **Supplementary Figures 4A and 4B**. The plot presented as **Supplementary Figure 4C** focuses on describing the overlap existing between the Louvain communities of each layer, in terms of CMS linked genes (i.e. that are either causal for the disease, present compound heterozygous mutations or copy number variants), for both severe and non-severe patients.

5) Along the same lines, it would be interesting to see the different interaction types displayed in Figures 3 and 4 to better appreciate the usefulness of the multilayer approach. Do the three layers contribute equally to the final modules?

We thank the reviewer for this remark. We have added the existing incident edges between nodes of the severe module as **Supplementary Figure 7**. This visualisation helps understanding the fact that the information producing the presented module comes from the interactome and reactome layers. As discussed in previous responses:

1. *LOXL3* does not present incident interactions with the other 14 severe-specific genes, which highlights its topological importance in the module: its inclusion has to do with incident interactions with other genes of the module (a total of 482 genes) that are not CMS linked genes (i.e. causal for the disease and /or presenting CNVs or compound heterozygous mutations).
2. Nodes inside the module are not present in the metabolomic layer. Still, it should be noted that this is not enough to rule out the importance of this layer on providing meaningful information. For instance, *ACOT2* gene, which we pinpoint as a severity candidate gene for Patient 3, presents important relationships through all layers with known CMS causal genes, as already stated in the response to comment no. 2.

Finally, a more conceptual question occurred to me while going through the manuscript: Do the authors consider mild disease as simply the absence of additional variants leading to a more severe outcome?

Or would they expect that the unique variants identified in the mild cases may have protective effects?

I understand that this question will likely have no general answer, but since the basic hypothesis of how variants modulate disease outcome would impact both the methodology for searching for disease modifying variants, as well as their interpretation, I believe a brief discussion would be instructive.

We thank the reviewer for this comment. Here, we present the first study of the CMS disease with a particular focus on secondary variants with the potential of increasing phenotypic severity, identifying specific mutations for both phenotypic subtypes as well as their functional relationship to already known causal genes of the disease.

As pointed out by the reviewer, one of the main future perspectives of the presented research should be on discerning whether mild-specific mutations could be alternative roles to the ones of severe-specific variants. As depicted in **Suppl. Figure 6**, the largest modules of the severe and non-severe groups share the same causal genes (*COL13A1*, *LAMA5*, *LAMB2*, *LRP4*, *PLEC* and *AGRN*), but differently mutated genes. This indicates a functional relationship between the two variant sets and therefore, with the same NMJ processes.

It is possible that severe-specific mutations affect genes more actively involved through the different layers, resulting in a higher impairment of the overall NMJ function. While for now we have targeted the hypothesis of severe-specific impairments increasing phenotypic

severity, we share the reviewer's vision on the potential of role of mild-specific mutations, being one of our main investigation lines for future studies on the disease

Although we agree that the whole section could be added to the main text, we currently opted out of doing so given the already considerable length of the manuscript, as we believe that, given the scope of our work, a major focus should be given to the multilayer network approach and the personalised analysis of the patient impairments. Nonetheless, we want to thank the reviewer for all the very constructive and helpful comments.

Reviewer #3 (Remarks to the Author):

Congenital myasthenia syndromes are a diverse group of neuromuscular junction disorders. CMS phenotypic severity varies dramatically, likely because they are caused by different mutations of same and/or different genes. Earlier studies were hampered by relatively small populations of CMS patients/samples. This multi-center study was able to group patients into severe disease group and no-sever group. Analysis of three main types of genetic variations (SNPs, CNVs, and compound heterozygous variants) did not render any SNPs as could be considered a unique cause of disease severity. The authors developed an algorithm to compare patient genomic data with the information provided by a biomedical knowledge multilayer network and were able to reveal functional relationships of new genetic modifiers responsible for different phenotypic severity levels. They explored phenotypes of knocking down one such modifier USH2A in zebrafish and demonstrated that the strategy seems to work.

The paper reports on a new workflow or strategy to explain why some patients with same mutation vary in disease severity, which could be applied to other rare disorders. I have only minor suggestions. First, the literature citation should be more accurate. For example, when the agrin-Lrp4-Musk pathway was described, a paper by Whicher, Philbin, and Aronson (2018) was cited. The paper was NOT on the pathway or NMJ at best; it was entitled “An Overview of the Impact of Rare Disease Characteristics on Research Methodology.”

We thank the reviewer for their appreciation of our work and agree on the need to improve the citation of the literature. Following the reviewer’s comment, we have removed the mentioned reference and changed it by the originally intended citation (Burden et al. 2013 *Cold Spring Harb. Perspect. Biol.*). We have checked all manuscript citations to ensure that they point to the correct references. We thank the reviewer for pointing out this error.

Recent reviews on NMJ formation/signaling should cited including Li et al., *Ann Rev Physiology* 2018.

We agree with the reviewer on the importance of the indicated review article and added it to the manuscript.

Second, the paper contains many typos. For example, line 54, This work showcase how... It should read “showcases”. Extensive proof reading is needed.

We have checked the manuscript for possible typing errors and corrected them accordingly.

REVIEWER COMMENTS

Reviewer #2 (Remarks to the Author):

The authors have addressed all my comments.

Reviewer #3 (Remarks to the Author):

The authors have done a reasonable job in revising the manuscript and addressing mine and concerns of the other two reviewers. I have no further comments.

Reviewer #4 (Remarks to the Author):

Authors performed additional studies to comply with the reviewer's concerns and suggestions. However, enigmas still remain.

1. The authors analyzed 20 CMS patients and tried to identify disease-modifying genes in approximately 20,000 genes. The small number of patients and the large number of genes can easily give rise to false positives that fit only to the authors' cohort but not to the other cohorts. The authors need to validate their finding using an external validation dataset.

2. Motor endplate is enriched in many proteins. Immunostaining of muscle proteins generally stains the motor endplate except for structural proteins that are known to function at the non-NMJ regions. GTEx is a database of tissue-specific gene expressions in human (<https://gtexportal.org/home/gene/USH2A>). According to GTEx, USH2A is highly expressed in the liver and testis, but its expression in skeletal muscle is quite low with a median TPM = 0.0086. Laser capture microdissection of mouse NMJ showed that Ush2a is not expressed in two reports (Ketterer et al. *Inves Ophthalmol Vis Sci* 51: 4589, 2010; Nazarian et al. *Physiol Genomics* 21: 70, 2005) or not included in the top 88 NMJ-enriched genes (McGeachie et al. *Mol Cell Neurosci* 30: 173, 2005). Similarly, in RiboTag mouse, the expression of Ush2a at the motor endplate was zero (Huang et al. *Front Mol Neurosci* 13: 154, 2020). It is hard to believe that USH2A is expressed at the NMJ.

3. The authors show WB and immunostaining of USH2A. Polyclonal antibodies frequently recognize nonspecific proteins. Considering low or zero expression of the Ush2a gene at the NMJ, the authors need to show the specificity of the antibody. In addition, WB and immunostaining provide essential information with readers of this article. The authors should show the data in Supplementary Figure.

POINT-TO-POINT RESPONSE TO REVIEWERS

Reviewer #2 (Remarks to the Author):

The authors have addressed all my comments.

We thank the reviewer for the very helpful comments, as we strongly believe that they enhanced the quality of the manuscript.

Reviewer #3 (Remarks to the Author):

The authors have done a reasonable job in revising the manuscript and addressing mine and concerns of the other two reviewers. I have no further comments.

We express our gratitude to the reviewer for their highly valuable comments. We firmly believe they have significantly improved the manuscript's quality.

Reviewer #4 (Remarks to the Author):

Authors performed additional studies to comply with the reviewer's concerns and suggestions. However, enigmas still remain.

*1) Motor endplate is enriched in many proteins. Immunostaining of muscle proteins generally stains the motor endplate except for structural proteins that are known to function at the non-NMJ regions. GTEx is a database of tissue-specific gene expressions in human (<https://gtexportal.org/home/gene/USH2A>). According to GTEx, *USH2A* is highly expressed in the liver and testis, but its expression in skeletal muscle is quite low with a median TPM = 0.0086.*

We agree with the reviewer on the importance of assessing the expression levels of *USH2A* in human skeletal muscle and understand their concerns regarding the expression levels reported by GTEx for this gene. The expression value reported by the reviewer is a quantification from bulk RNA-seq of skeletal muscle tissue. We believe that this particular quantification may not effectively serve as a reliable indicator to definitively ascertain or dismiss the expression of *USH2A* at the NMJ or its potential involvement in CMS severity.

Undoubtedly, bulk RNA-seq studies have played a pivotal role in unravelling the biology of skeletal muscle; however, such insights are hampered by profiling challenges that may lead to the averaging of expression across diverse cell populations, a concern that becomes even more pronounced for the NMJ (Williams et al. *Frontiers in genetics*, 2022, doi:10.3389/fgene.2022.835099). As a point of reference, NMJ nuclei constitute a mere 0.8% of the total myonuclei found in the adult mouse tibialis anterior (Petraný et al. *Nature communications*, 2020, doi:10.1038/s41467-020-20063-w). Thus, obtaining insights into rare

disease mechanisms that specifically affect certain cell populations, such as the NMJ, proves to be exceptionally intricate.

By further examining the GTEx summary statistics in bulk skeletal muscle (see **Table 1** below), we observed overall low expression levels of previously described CMS causal genes, which also appear in our severe-specific module (*LAMA5*, *LAMB2*, *AGRN*, *LRP4*, *PLEC* and *COL13A1*). Additionally, we examined GTEx single-cell RNA-seq expression levels of CMS causal genes in a specific skeletal muscle cell type related to NMJ (labelled as ‘Skeletal Muscle, Myocyte NMJ-rich’), as well as the median TPM values in the Tibial Nerve (Peripheral Nervous System) tissue (see **Table 1** below). Interestingly, the median TPM values of the known CMS causal genes and *USH2A* exhibit higher rankings within bulk tibial nerve when contrasted with those of bulk skeletal muscle. The expression levels of *PLEC* are particularly high. This gene encodes plectin, a giant protein that plays a key role in upholding the mechanical stability of various tissues.

Gene	scRNA-seq of Myocyte NMJ-Rich cells: detected in cells/sample size (%)	Bulk RNA-seq of Skeletal muscle: median TPM (overall ranking)	Bulk RNA-seq of top 3 tissues: Tissue (TPM)	Bulk RNA-seq of Tibial nerve: median TPM (overall ranking)	Link
LAMA5	4/95 (4.21%)	10.60 (41/54)	Colon Sigmoid (175.2) Artery Tibial (171.8) Brain Cerebellum (127.8)	76.63 (19/54)	https://www.gtexportal.org/home/gene/LAMA5
LAMB2	2/95 (2.11%)	51.45 (36/54).	Artery Aorta (352.4) Ovary (317.6) Artery Tibial (293.9)	11.3 (13/54)	https://www.gtexportal.org/home/gene/LAMB2
LRP4	56 / 134 (41.79%)	3.977 (31/54)	Skin, Sun-exposed (55.88) Skin, Not Sun-exposed (55.22) Brain Caudate (38.70)	11.91 (14/54)	https://www.gtexportal.org/home/gene/LRP4
PLEC	2/95 (2.11%)	151.2 (3/54)	Nerve Tibial (241.5) Cells - Cultured fibroblasts (212.1) Muscle - Skeletal (151.2).	241.5 (1/54)	https://www.gtexportal.org/home/gene/PLEC

AGRN	Not Reported Myocyte, Skeletal Muscle: 30 / 20772 (0.14 %)	2.403 (53/54)	Thyroid (97.41) Kidney Medulla (74.23) Cells - EBV - Transformed lymphocytes (57.03)	11.91 (23/54)	https://www.gtexportal.org/home/gene/AGRN
COL13A1	35/95 (36.84%)	0.5513 (41/54).	Brain Cerebellar Hemisphere (83.94) Brain Cerebellum (80.74) Artery Aorta (30.62)	0.5877 (38/54)	https://www.gtexportal.org/home/gene/COL13A1
USH2A	8/95 (8.42%)	0.0086 (52/54)	Testis (1.215) Liver (0.8047) Brain Cerebellar Hemisphere (0.22)	0.08234 (10/54)	https://www.gtexportal.org/home/gene/USH2A

Table 1 (Reply). GTEx summary statistics of bulk RNA-seq and single-cell RNA-seq expression data for known CMS causal genes in the severe-specific module and *USH2A*. In bulk RNA-seq data, *PLEC* is the only CMS causal gene expressed in skeletal muscle with a notably high median TPM. Conversely, known CMS causal genes, such as *AGRN*, *COL13A1* and the laminin coding genes *LAMB2* and *LAMA5*, display extremely low expression levels in skeletal muscle.

In addition to the explored GTEx information, another relevant resource reporting single cell RNA-seq data from NMJ-specific cell populations is available in the Myoatlas database (<https://research.cchmc.org/myoatlas/>) (Petraný et al. *Nat Commun*, 2020, doi: 10.1038/s41467-020-20063-w). By analysing this resource, we observed a trend similar to that of GTEx (i.e., low expression of CMS causal genes, comparable to *USH2A* in the specific NMJ-specific cell populations) (**Figure 1 of the Reply**).

In conclusion, we acknowledge that further research is needed to investigate the relationship between the impact of candidate modifier genes on disease severity and their expression levels. Thus, we do not advise to draw any conclusions about the association of genes with CMS based solely on expression levels. For this reason, we believe that focused investigations utilising protein assays (such as Western blotting and ELISA) in animal models, such as zebrafish and mice as in our own study, allow obtaining indispensable biomedical knowledge.

Figure 1 (Reply). Expression levels of typical NMJ markers (*Musk*, *Ache*, *Lrp4*, *Chrne*), as well as relevant CMS causal genes (*Agrn*, *Rapsn*, *Dok7*), and *Ush2a* in NMJ cell type from single-cell data from the Myoatlas database (<https://research.cchmc.org/myoatlas/>) (Petraný et al. *Nat Commun*, 2020, doi: 10.1038/s41467-020-20063-w).

2) Laser capture microdissection of mouse NMJ showed that *Ush2a* is not expressed in two reports (Ketterer et al. *Inves Ophthalmol Vis Sci* 51: 4589, 2010; Nazarian et al. *Physiol Genomics* 21: 70, 2005) or not included in the top 88 NMJ-enriched genes (McGeachie et al. *Mol Cell Neurosci* 30: 173, 2005). Similarly, in RiboTag mouse, the expression of *Ush2a* at the motor endplate was zero (Huang et al. *Front Mol Neurosci* 13: 154, 2020). It is hard to believe that *USH2A* is expressed at the NMJ.

We thank the reviewer for providing evidence of the difficulties in assessing the expression of *Ush2a* at the NMJ. This remark motivated us to delve into the literature provided by the reviewer and identify additional resources supporting the potential role of *Ush2a* at the NMJ.

The first mentioned article (*Ketterer et al. Invest Ophthalmol Vis Sci 51: 4589, 2010*) provides highly valuable information on the differential expression between Tibialis anterior synapse and the Tibialis anterior fiber (**Table S2 of the provided article**). Indeed, *USH2A* is not differentially expressed, but this is also the case for a good number of CMS causal genes. Of the causal genes appearing in our module, *LAMA5*, *PLEC* and *COL13A1* do not appear with a FC > 2 for the relevant comparative analysis (**Tibialis anterior synapse vs Tibialis anterior fiber**). Similarly, other known CMS causal genes are not reported as differentially expressed either (*MUSK*, *DOK7*, *RAPSN*, *RPH3A*, *CHAT*, *PREPL*, *SLC5A7*, *VAMP1*, *UNC13B* among others). In this sense, it is evident that the presence of differential expression at the Tibialis anterior synapse is not a crucial factor in influencing the manifestations of the CMS disease.

The second article (*McGeachie et al. Mol Cell Neurosci 30: 173, 2005*) provides interesting information on the specific transcriptome of the postsynaptic elements. Importantly, none of the causal CMS genes appearing in the identified module are reported among the top-88 gene list pointed by the reviewer (**Table 2A of the provided article**). In general, important information might be missing as a result of the described muscle fibre isolation. In this particular publication, muscle fibres are isolated and purified, while terminal Schwann Cells, as well as other cell components of the NMJ, are removed and therefore not taken into account in the study. The presynaptic location is a particularly important aspect to consider for CMS causal genes, as well as for *USH2A*, as a recent study reports the involvement of the gene in normal vibration sensing, being present in terminal Schwann Cells of Messnier corpuscles (*Schwaller et al., Nat. Neurosci. 2021*). Moreover, previous studies also comment on the relatively limited literature focusing on the role of Schwann Cells at the NMJ (*Santosa et al., J Neurosci Res, 2018*).

The third article (*Huang et al. Front Mol Neurosci 13: 154, 2020*) provides information on the transcriptome at the motor endplate. Here, we inspected the reported expression for all causal CMS genes. In general, there is a significant absence of expression signal for causal CMS genes interacting at the presynaptic level. The genes *CHAT*, *SLC5A7*, *RPH3A* and *SYT2* present zero TPM expression, while *SNAP25* has virtually zero expression. These results clearly indicate that there may be a loss on the signal coming from the presynaptic component of the NMJ.

Furthermore, recent literature supports the potential role of *USH2A* at the NMJ level. In addition to the already mentioned publication from Schwaller et al. reporting the role of the protein Messnier corpuscle Schwann Cells, we place a strong emphasis on the study of *Yoshida et al. Stem Cell Rep., 2015* (<https://doi.org/10.1016/j.stemcr.2015.02.010>), which focuses on the potential treatments of Spinal Muscular Atrophy, a disease that is triggered at

the presynaptic level and that has high overlap with the clinical manifestations of CMS. Particularly, this publication reports how Valproic Acid, a potential treatment of the disease that enhances AChR Clustering (the type of treatments suggested in our manuscript as a result of the identified severity modifiers) upregulates USH2A expression at the NMJ, among other genes, including *PPFIBP2*, which we identified among the specific genes presenting compound heterozygous variants for patient 3 (**Functional consequences of variants in the severe-specific module, Lines 382-389**). Given its significance for our study, we added this reference to the revised manuscript (**Experimental validation of USH2A involvement at the NMJ, Lines 431-434**: “we hypothesised that knockdown of *USH2A* expression alone may cause detectable NMJ impairments, as a previous study on Spinal Muscular Atrophy (Yoshida et al., 2015) reported *USH2A* among the genes whose expression was upregulated upon treatment with valproic acid, an AChR clustering enhancer”). We have a firm conviction that the literature provides substantial evidence supporting the existence and a possible pivotal role of usherin protein at the NMJ level in disease.

3) The authors analyzed 20 CMS patients and tried to identify disease-modifying genes in approximately 20,000 genes. The small number of patients and the large number of genes can easily give rise to false positives that fit only to the authors' cohort but not to the other cohorts. The authors need to validate their finding using an external validation dataset.

We appreciate the reviewer's comment and acknowledge their concern regarding the generalizability of our conclusions to other CMS cohorts. It is important to note that, to the best of our knowledge, well-characterised cohorts for this particular rare condition are scarce. Indeed, our study stands as a culmination of **nearly a decade of international collaborative research efforts dedicated to the deep phenotypic and genomic characterisation of CMS**. Moreover, our cohort's uniqueness is further highlighted by its specific ethnic composition, a factor that was meticulously considered in the genomic analysis as reported in the manuscript. We expect that a comparably extensive effort will be necessary to establish an additional similar CMS cohort in the future.

As for the analytical procedure that led us to identify disease-modifying genes, we would like to provide some clarifications. As outlined in the manuscript (**Lines 133-137**), an extensive testing of a wide array of mutation types (refer to "Segregation analyses" in the Supplemental Information, Figure S1, and Supplemental Table 2) revealed that only Copy Number Variations (CNVs) and Compound Heterozygous Variants (CHVs) exhibited a partial yet promising indication of differentiation between severity groups, along with a significant functional enrichment of known CMS-related pathways. This step is essential to (1) rule out the possibility that genetic heterogeneity alone accounts for the diverse clinical manifestations observed, and (2) concentrate on identifying specific processes underlying disease consequences, in line with established best practices for investigating modifier genes (Génin et al. *Human Genetics*, 2008, doi:10.1007/s00439-008-0560-2). Thus, our analysis focused specifically on CNVs and CHVs, which significantly reduced the genomic search space. For instance, the pool of severe-specific mutations include 89 genes presenting CHVs

and 10 genes presenting CNVs. We provide the complete lists of the genes presenting these mutations over the severe and not-severe patients in the following table at Github:

<https://github.com/ikernunezca/CMS/blob/master/data/InputGenes/AllGenes.csv>

We would like to emphasise that our methodology, centred around multilayer networks, offers the significant advantage of enabling a personalised functional analysis of the molecular determinants of rare disease severity. Specifically, it allows identifying gene modules that, through their involvement in functions linked to causal processes of CMS, result in comparable levels of severity when subject to patient-specific alterations. Thus, should we incorporate more patients from an external cohort into the analysis, we would expect to identify either the very same genes, if mutated, or other genes belonging to the same or functionally close modules.

Our methodology for the identification of genetic modifiers of CMS severity helps accelerate research in a scenario critically hindered by the scarcity of external cohorts for validation, which represents one of the major challenges in rare disease research. Nevertheless, we put a lot of effort in providing complementary and enriching experimental validation in two animal models (zebrafish and mouse), which not only corroborated our computational findings but also enabled us to enhance our understanding of the relevance of usherin in CMS.

4. The authors show WB and immunostaining of USH2A. Polyclonal antibodies frequently recognize nonspecific proteins. Considering low or zero expression of the Ush2a gene at the NMJ, the authors need to show the specificity of the antibody. In addition, WB and immunostaining provide essential information with readers of this article. The authors should show the data in Supplementary Figure.

As the reviewer highlighted we did use a polyclonal antibody for the labelling. We selected this antibody provided the scarcity of commercially available antibodies available for the usherin protein. Following the reviewer's petition, we have now demonstrated that this antibody labels synapses in the brain, as predicted (**Figure 2 of this reply, Supplementary Figure 12 in the manuscript**). In addition, we show that there is no non-specific binding of the secondary antibody.

Figure 2 (Reply) and Supplemental Figure 12 (Manuscript). Detection of Usherin in different tissues. **(A)** Western blot performed on 10-week-old mice detected a single band of the expected size in the spinal cord, liver, and tibialis anterior muscle, which is compatible with the ubiquitous expression of the protein in multiple tissues. Unfortunately, we were unable to detect the protein in the brain (Rb polyclonal IgG, FabGennix). **(B)** Using a directly conjugated Anti-Usherin-FITC antibody (Rb polyclonal, FabGennix) we were able to detect by immunofluorescence USH2A in mouse brain and at some neuromuscular junctions (α -Bungarotoxin) in mouse soleus muscle.

We agree with the reviewer's suggestion regarding the potential interest of readers in this information. Thus, we incorporated it into the manuscript's discussion as **Supplementary Figure 12 (Lines 623-630)**: ‘Previous studies have commented on USH2A presence on the basement membranes of perineurium nerve fibers (Pearsall et al., 2002; Schwaller et al., 2021), however, further studies in a mammalian model and/or using zebrafish mutants rather than transient knockdown will be required to determine the presence of USH2a at the NMJ, and whether loss of USH2a alone can impact NMJ signalling or whether co-occurrence with CHRNE CMS is required. In this sense, we report evidence of USH2A presence at the tibialis anterior muscle in 10-week-old mice (**Supplementary Figure 12A**) and the soleus muscle (**Supplementary Figure 12B**) (**Methods**)’. We report on the methodology of this experiment in the Methods Section, accordingly (**Lines 853-870**).

REVIEWERS' COMMENTS

Reviewer #4 (Remarks to the Author):

The authors have addressed my concerns on the paucity of synaptic expression of Ush2a and anti-Ush2a antibody. I understand that it is almost impossible to draw a biologically definite conclusion with a small number of samples using a large number of features. This is inevitable especially in analyzing the human genome. A lot of false positive conclusions have been reported in our scientific community, but many publications successfully disclosed hidden scenarios in medicine and biology. I still have a concern that the conclusions presented in this manuscript may be applicable only to their dataset. However, if we stick to this concern, no similar reports can be published in our community, which should retard the advancement in medicine and biology.

Reviewer #5 (Remarks to the Author):

I have been asked, as an expert in modeling neuromuscular disorders in zebrafish, to comment on the strength of the zebrafish experiments discussed in the submitted manuscript. Please see below:

- This study uses a single morpholino to knockdown *ush2a* in zebrafish. The use of morpholinos is rather dated and without the right controls can produce unreliable phenotypes. There are lots of better ways of knocking down or knocking out gene function in zebrafish, the gold standard is making a mutant line but this can be time consuming. However, a CRISPR-mediated multiguide approach can be used to target biallelic changes in the F0, making a cleaner faster way to knock down gene function. This works very nicely when a germline stable mutant is not present. See published work from Jason Rehel's group (Kroll et al (2021) A simple and effective F0 knockout method for rapid screening of behaviour and other complex phenotypes eLife). However, for *ush2a* there are mutants available!! (Hans et al 2018 Knockout of *ush2a* gene in zebrafish causes hearing impairment and late onset rod-cone dystrophy. Hum Genet, Dona et al 2018 Usherin defects lead to early-onset retinal dysfunction in zebrafish Exp Eye Res). If I decided to go down the morpholino route, because of time restraints, I would look to see if there was already a published *ush2a* MO that could be used. This would save on all the validation experiments that are necessary to prove specificity of new MOs. According to ZFIN, the zebrafish database, there are 7 published MOs mainly from 2 papers (Ebermann et al 2010 and Dulla et al 2021), checking the published MO sequences, none are the same as Núñez-Carpintero et al's. Therefore, this team have decided to design and characterise a new unpublished MO.

Since they have used a new uncharacterised MO they need to prove that their MO is specific. They claim this is provided through RT PCR of morphants, it's not clear from this experiment where the primers have been designed. Classically for a splice morpholino one would PCR using primers that flank the intron and primers that are within the intron. This would provide evidence of an intronic inclusion. What would be the repercussions of an intronic inclusion? Premature stop? This is not discussed. A schematic showing the position of the primers would be helpful. It seems to me that exposure of this gel is low, compare the wildtype band with the wildtype band at 2dpf in 'A'. OK, so lets say we are seeing RNA mediated decay, it would certainly make it easier to read if we knew the specifics of the experiment.

Some years ago there were guidelines published for the use of MOs in zebrafish knockdown experiments (Eisen JS, Smith JC. Controlling morpholino experiments: don't stop making antisense. *Development*. 2008) . These have not been followed in the Núñez-Carpintero manuscript. At the very least I would expect the researchers to use a second non-overlapping morpholino to prove that a similar affect is seen with a different MO. 18ng of morpholino is extremely high in my opinion and usually causes toxicity. For movement defects it would have been better to look at more free-swimming stages of development such as 4 or 5dpf. There is usually a lot of variability at the younger stages and this is reflected in the presented graphs. Núñez-Carpintero et al look at 2dpf, at this stage embryos are only just starting to break free of the chorion. Why restrict your study to 2dpf? Morpholinos often cause developmental delay and I wonder if some of the defects reported are due to this, how do they look later? In supplementary fig10D, images of controls and morphants are shown. What is noticeable is that neither controls or morphants have pigment. If PTU or a pigment mutant line was used, then this needs to be stated, it's not.

Núñez-Carpintero et al perform staining of synaptic vesicles and AChR to determine if NMJ clustering is affected in Morphants. It's almost impossible to say anything conclusive from the images provided. There needs to be more annotation on figures and at least an inset or zoomed image showing increased vesicle clustering or more SV2 on fast muscle fibres. Incidentally, there is no evidence from the image to say whether the clustering is on fast or slow muscle. The image is a z-stack projection which will include both superficial slow fibres and the deeper fast. These claims are unsubstantiated. How were these counted? using software or a blinded researcher? I see a subtle change but I also see a noticeable change in the myotome chevron shape, they are less angled. Are the suggested changes specific to neurons or due to truncation of the myotome? A second morpholino is needed here, although a mutant would be better.

The main point of the zebrafish work is to provide evidence that mutations in a detected modifier gene (*ush2a*) can change the severity of a phenotype from a causal gene mutation (*CHRNE*). The morpholino experiment does not answer this in the present state. To prove that *CHRNE* and *ush2a* are working in the same genetic pathway, the best experiment would be to inject sub-phenotypic levels of morpholinos for both genes, which would hopefully show an NMJ phenotype in the doubles only.

Conclusion:

The fish experiments do nothing to support the data being presented in this paper. It feels like a last-minute add-on which hasn't been thought through. If the authors want to use zebrafish to support their in-depth multi-omics approach to identify potential modifiers, then they need to come up with a more eloquent assay, as suggested earlier. Further to this, I would suggest contacting one of the other groups for some *ush2a* mutants to analyse NMJs or make a *chrne* mutant and see if the phenotype is exacerbated with *ush2a* knockdown (use 2 non-overlapping MOS). In the current state, and based on the zebrafish work, I do not recommend the publication of this manuscript.

Point-to-point response to the Reviewers' comments

Reviewer #4 (Remarks to the Author):

The authors have addressed my concerns on the paucity of synaptic expression of Ush2a and anti-Ush2a antibody. I understand that it is almost impossible to draw a biologically definite conclusion with a small number of samples using a large number of features. This is inevitable especially in analyzing the human genome. A lot of false positive conclusions have been reported in our scientific community, but many publications successfully disclosed hidden scenarios in medicine and biology. I still have a concern that the conclusions presented in this manuscript may be applicable only to their dataset. However, if we stick to this concern, no similar reports can be published in our community, which should retard the advancement in medicine and biology.

We thank the reviewer for the comments. We understand the concern on the potential extrapolation of the manuscript conclusions to other cohorts of Congenital Myasthenic Syndromes. The lack of validation cohorts with a similar level of detail is a general problem in rare disease research.

We believe it is important to be cautious when asserting claims about the potential results of similar analyses in other CMS cohorts. Although we may expect severe patients to present damaging mutations, affecting genes functionally related to those observed in our cohort, this could only be confirmed with dedicated studies on other CMS cohorts. In this sense, our study not only suggests a set of new target genes that may play a role in CMS severity but also establishes a basis for uncovering additional potential contributors to the manifestations of the disease in the future.

We agree with the reviewer on the importance of this discussion and would like to thank her/him again for raising it.

Reviewer #5 (Remarks to the Author):

This study uses a single morpholino to knockdown *ush2a* in zebrafish. The use of morpholinos is rather dated and without the right controls can produce unreliable phenotypes. There are lots of better ways of knocking down or knocking out gene function in zebrafish, the gold standard is making a mutant line but this can be time consuming. However, a CRISPR-mediated multiguide approach can be used to target biallelic changes in the F0, making a cleaner faster way to knock down gene function. This works very nicely when a germline stable mutant is not present. See published work from Jason Rehel's group (Kroll et al (2021) A simple and effective F0 knockout method for rapid screening of behaviour and other complex phenotypes eLife). However, for *ush2a* there are mutants available!! (Hans et al 2018 Knockout of *ush2a* gene in zebrafish causes hearing impairment and late onset rod-cone dystrophy. Hum Genet, Dona et al 2018 Usherin defects lead to early-onset retinal dysfunction in zebrafish Exp Eye

Res). If I decided to go down the morpholino route, because of time restraints, I would look to see if there was already a published *ush2a* MO that could be used. This would save on all the validation experiments that are necessary to prove specificity of new MOs. According to ZFIN, the zebrafish database, there are 7 published MOs mainly from 2 papers (Ebermann et al 2010 and Dulla et al 2021), checking the published MO sequences, none are the same as Núñez-Carpintero et al's. Therefore, this team have decided to design and characterise a new unpublished MO.

Since they have used a new uncharacterised MO they need to prove that their MO is specific. They claim this is provided through RT PCR of morphants, it's not clear from this experiment where the primers have been designed. Classically for a splice morpholino one would PCR using primers that flank the intron and primers that are within the intron. This would provide evidence of an intronic inclusion. What would be the repercussions of an intronic inclusion? Premature stop? This is not discussed. A schematic showing the position of the primers would be helpful. It seems to me that exposure of this gel is low, compare the wildtype band with the wildtype band at 2dpf in 'A'. OK, so lets say we are seeing RNA mediated decay, it would certainly make it easier to read if we new the specifics of the experiment.

Some years ago there were guidelines published for the use of MOs in zebrafish knockdown experiments (Eisen JS, Smith JC. Controlling morpholino experiments: don't stop making antisense. *Development*. 2008) . These have not been followed in the Núñez-Carpintero manuscript. At the very least I would expect the researchers to use a second non-overlapping morpholino to prove that a similar affect is seen with a different MO. 18ng of morpholino is extremely high in my opinion and usually causes toxicity. For movement defects it would have been better to look at more free-swimming stages of development such as 4 or 5dpf. There is usually a lot of variability at the younger stages and this is reflected in the presented graphs. Núñez-Carpintero et al look at 2dpf, at this stage embryos are only just starting to break free of the chorion. Why restrict your study to 2dpf? Morpholinos often cause developmental delay and I wonder if some of the defects reported are due to this, how do they look later? In supplementary fig10D, images of controls and morphants are shown. What is noticeable is that neither controls or morphants have pigment. If PTU or a pigment mutant line was used, then this needs to be stated, it's not.

Núñez-Carpintero et al perform staining of synaptic vesicles and AChR to determine if NMJ clustering is affected in Morphants. It's almost impossible to say anything conclusive from the images provided. There needs to be more annotation on figures and at least an inset or zoomed image showing increased vesicle clustering or more SV2 on fast muscle fibres. Incidentally, there is no evidence from the image to say whether the clustering is on fast or slow muscle. The image is a z-stack projection which will include both superficial slow fibres and the deeper fast. These claims are unsubstantiated. How were these counted? using software or a blinded researcher? I see a subtle change but I also see a noticeable change in the myotome chevron shape, they are less angled. Are the suggested changes specific to neurons or due to truncation of the myotome? A second morpholino is needed here, although a mutant would be better.

The main point of the zebrafish work is to provide evidence that mutations in a detected modifier gene (*ush2a*) can change the severity of a phenotype from a causal gene mutation (*CHRNE*). The morpholino experiment does not answer this in the present state. To prove that *CHRNE* and *ush2a* are working in the same genetic pathway, the best experiment would be to inject sub-phenotypic levels of morpholinos for both genes, which would hopefully show an NMJ phenotype in the doubles only.

We thank the reviewer for the comments. Our experiments with zebrafish follow the same methodology presented in previous analysis (O'Connor 2018, 19). These experiments, performed during the COVID19 pandemic, have required 2 years to achieve the observed results that confirm the presence of the USH2A protein at the NMJ of zebrafish and mouse. We show specific neuromuscular phenotypes that are unlikely unspecific effects of the morpholino. Moreover, p53 morpholino co-injections did not alter the results supporting our claim that the effects are specific and due to the knockdown of *ush2a*. Other models such as crisprants or crspr-generated zf lines have limitations, disadvantages and drawbacks, as the reviewer is certainly aware of.

The reviewer states that the main point of the zebrafish work is '*to provide evidence that mutations in a detected modifier gene (ush2a) can change the severity of a phenotype from a causal gene mutation (CHRNE)*'. We never make such a claim and we believe that, prior to analysing if that might be the case, it is imperative to establish evidence demonstrating that the mutation of USH2A can indeed induce effects in skeletal muscle, as there is no previous studies reporting such effects. Indeed, models targeting USH2A have been generated focusing on the study of Usher Syndrome, not providing evidence about implications in skeletal muscle. While the neuromuscular junction in zebrafish larvae is a good model to study basic and disease mechanisms, the severity of conditions such as CMS is not resembling human disease as seen for a variety of known CMS-causing genes.

However, we do acknowledge that the ultimate objective of any future experimental work should aim to discern if the CMS modifying effect coming from USH2A mutations requires the previous presence of the *CHRNE* mutations. In this sense, we comment about this requirement in the Discussion: '*further studies in a mammalian model and/or using zebrafish mutants rather than transient knockdown will be required to determine the presence of USH2a at the NMJ, and whether loss of USH2a alone can impact NMJ signalling or whether co-occurrence with CHRNE CMS is required. Additional functional work is also required to ascertain the importance of other potential modifiers identified in this study. Particularly, a prospective analysis on the potential NMJ involvement of the unique variants detected for the non-severe group could be of special interest for the study of CMS, potentially discerning their functional relationship to causal CMS genes*'.

Based on all these considerations, we decided to remove the experiments on zebrafish from the manuscript.